# Distinct mechanisms of replication stress induced by oncogenic RAS and cyclin E1 converge on R-loop-dependent fork reversal

Anna Oravetzova [1,2,9], Marketa Dvorakova [3,9], Anca-Irina Mihai [3,9], Martin Andrs [3,4], Margarita Sobol [5,6], Anton Zuev [1,2], Kaustubh Shukla [1], Barbora Boleslavska [1,8], Vinicio Rosano [3], Christiane König [3], Jiri Prokes [3], Hana Hanzlikova [5,7], Libor Macurek [1], Jana Dobrovolna [1,4] & Pavel Janscak [1,3] ✉

Activated oncogenes elicit genomic instability by inducing DNA replication stress. Here we show that replication fork reversal and chromosome mis-segregation induced by oncogenic RAS (HRASV12) or cyclin E1 overexpression are largely caused by co-transcriptional RNA:DNA hybrids (R-loops) formed during S-phase. Furthermore, we demonstrate that replication stress induced by HRASV12, but not cyclin E1, is driven by reactive oxygen species (ROS) in a manner dependent on the replisome-associated ROS sensor peroxiredoxin 2 (PRDX2) and is linked to PRDX2-mediated release of the fork acceleration factor TIMELESS from the replisome. Inhibition of fork reversal in cells over-expressing HRASV12 or cyclin E1 induces unrestrained DNA synthesis mediated by the MUS81 endonuclease and the primase-polymerase PRIMPOL, thereby promoting proper chromosome segregation in mitosis. These results establish PRIMPOL repriming as part of the MUS81-dependent replication restart mechanism that operates at sites of R-loop-mediated transcription-replication conflicts to maintain genomic stability. Furthermore, our data indicate that, despite their protective role during S-phase, persistent reversed forks impair chromosome segregation in mitosis, potentially leading to DNA breaks and chromosomal rearrangements.

Cancer is caused by genetic alterations that activate oncogenes and inactivate tumor suppressor genes. Oncogenes are known to induce stalling and collapse of DNA replication forks in precancerous lesions, leading to persistent activation of the DNA damage response machinery[1–4]. This phenomenon, known as oncogene-induced replication stress creates a selective pressure for the outgrowth of malignant clones carrying mutations in the tumor suppressor proteins such as p53, which inactivate the DNA damage checkpoints, allowing the cancer to develop[5]. One of the prototypical replication stress-inducing oncogenes is oncogenic RAS[3,6]. The RAS family consists of

[1]Laboratory of Cancer Cell Biology, Institute of Molecular Genetics of the Czech Academy of Sciences, Vídeňská 1083, Prague 4, Czech Republic. [2]Faculty of Science, Charles University in Prague, Albertov 6, Prague 2, Czech Republic. [3]Institute of Molecular Cancer Research, University of Zurich, Strickhofstrasse 40a, Zurich, Switzerland. [4]Department of Genome Biology, Institute of Experimental Medicine of the Czech Academy of Sciences, Vídeňská 1083, Prague 4, Czech Republic. [5]Laboratory of Genome Dynamics, Institute of Molecular Genetics of the Czech Academy of Sciences, Vídeňská 1083, Prague 4, Czech Republic. [6]Department of Cell Biology, Faculty of Science, Charles University, BIOCEV, Průmyslová 595, Vestec, Czech Republic. [7]Institute of Animal Pathology, Vetsuisse Faculty, University of Bern, Länggassstrasse 122, Bern, Switzerland. [8]Present address: DIANA Biotechnologies, a.s., Průmyslová 596, Vestec, Czech Republic. [9]These authors contributed equally: Anna Oravetzova, Marketa Dvorakova, Anca-Irina Mihai. ✉e-mail: pjanscak@imcr.uzh.ch

three proto-oncogenes - *KRAS*, *NRAS* and *HRAS* - which function as membrane-associated GTPase signal transducers that, upon GTP binding, activate multiple mitogenic pathways including RAF/MEK/ERK and PI3K/AKT[7,8]. RAS is turned into an oncogene by a mutation in the conserved codons 12, 13, or 61, which impairs intrinsic GTP hydrolysis or binding to GTPase-activating proteins, making RAS constitutively active. Such *KRAS* mutations are most commonly found in colorectal cancer (33%), pancreatic cancer (57%), and non-small-cell lung cancer, while *HRAS* mutations are associated with skin and head and neck cancers, and *NRAS* mutations are common in hematopoietic malignancies[7,8]. Evidence suggests that replication stress induced by oncogenic RAS arises from increased global transcription driven by elevated expression of the TATA-box binding protein (TBP), leading to the accumulation of R-loops[9]. These three-stranded nucleic acid structures, composed of an RNA:DNA hybrid and a single-stranded DNA (ssDNA) loop, can form co-transcriptionally by invasion of the nascent transcript into the underwound DNA duplex behind the transcription complex, and act as a strong barrier to replication fork progression[10]. However, a recent study suggested that, rather than inducing R-loop formation, the transcriptional burst triggered by oncogenic RAS leads to H3K27me3- and HP1-associated chromatin compaction, thereby resulting in replication fork stalling and cell death[11]. Moreover, it should be noted that sustained mitogenic stimulation by oncogenic RAS increases the intracellular concentration of reactive oxygen species (ROS)[12,13], which are known to induce R-loop-mediated fork stalling by promoting R-loop formation at sites of transcription-replication conflicts (TRCs) in a manner dependent on the replisome-associated ROS sensor peroxiredoxin 2 (PRDX2)[14]. Thus, the molecular mechanism underlying DNA replication stress induced by this oncogene remains elusive.

Another prototypical oncogene that induces DNA replication stress is cyclin E1, the product of the *CCNE1* gene[1,2]. Cyclin E1, in complex with CDK2, normally drives the progression of the cell cycle from the G1 phase to the S phase by phosphorylating multiple proteins, including RB, leading to the expression of various genes required for DNA replication[15,16]. However, cyclin E1 is frequently overexpressed in premalignant lesions and cancers such as ovarian tumors or uterine carcinomas due to gene amplification[15,17], which induces DNA replication stress[2]. Accumulating evidence suggests that cyclin E1 overexpression induces DNA replication stress by enhancing replication origin firing[18,19]. Cyclin E1 overexpression dramatically shortens the length of G1 phase, resulting in firing of intragenic origins that are normally inactivated by transcription in G1[19]. The firing of these oncogene-induced origins gives rise to transcription-dependent chromosome breakage at sites that coincide with chromosomal rearrangement breakpoints in a large cohort of cancers[19]. Thus, cyclin E1-induced replication stress is caused by aberrant firing of intragenic origins and subsequent head-on TRCs, which can lead to fork collapse and genomic instability. Although head-on TRCs are known to promote R-loop formation[20-23], particularly when replication fork progression is slowed due to impairment of the replication machinery[14,24], it remains to be determined whether R-loops are a major cause of DNA replication stress induced by cyclin E1 overexpression.

R-loops arrest replication fork progression by inducing replication fork reversal[14,25]. This fork remodeling process, mediated by the RAD51 recombinase and the DNA translocases ZRANB3, HLTF, and SMARCAL1, results in the pairing of nascent DNA strands to form a four-way DNA structure, where the newly formed DNA arm is covered by BRCA2-stabilized RAD51 filament to prevent nucleolytic resection[25]. Although overexpression of oncogenes such as cyclin E1 or CDC25A has been shown to induce fork reversal[26], the role of R-loops in these processes has not yet been investigated.

R-loop-stalled forks can be reactivated in a multistep process triggered by the action of two DNA helicases, namely RECQ1 and RECQ5[27]. RECQ1 helicase mediates reverse branch migration to eliminate reversed forks while RECQ5 helicase removes RAD51 from the stalled fork to prevent further fork reversal and to promote fork cleavage by the MUS81-EME1 endonuclease[27-29]. This initiates the replication restart process that additionally requires the strand-annealing factor RAD52, the DNA ligase IV (LIG4)/XRCC4 complex, the RNA polymerase II (RNAPII) elongation factor ELL, and the non-catalytic subunit of DNA polymerase delta POLD3[27]. Defects in this DNA repair pathway lead to genomic instability due to impaired chromosome segregation in mitosis caused by the presence of regions of incompletely replicated DNA[27]. However, its importance for DNA replication in oncogene-expressing cells remains to be determined.

Here, we show that the overexpression of HRASV12 or cyclin E1 induces R-loop formation specifically in the S-phase of the cell cycle and that R-loops are the primary cause of replication fork reversal and chromosome mis-segregation induced by these oncogenes. Furthermore, we demonstrate that R-loop-mediated replication stress induced by HRASV12, but not by cyclin E1, is largely driven by ROS in a manner dependent on PRDX2 and is linked to PRDX2-mediated release of the fork acceleration factor TIMELESS from the replication fork. Suppression of fork reversal in cells overexpressing HRASV12 or cyclin E1 restores normal replication fork progression in a manner dependent on the primase-polymerase PRIMPOL and proteins that mediate the reactivation of R-loop-stalled forks. Our data suggest that persistent reversed forks can give rise to genomic instability if not restarted before the onset of mitosis. In addition, these findings identify the MUS81-PRIMPOL axis as a potential therapeutic target for cancers expressing oncogenes that induce transcription-dependent replication stress.

## Results

### HRASV12- and cyclin E1-induced replication stress is mainly caused by R-loops

To study the molecular mechanisms underlying oncogene-induced replication stress, we used pBABEneo-based retroviral vectors for overexpression of HRASV12 and cyclin E1, respectively, in human cells (Supplementary Fig. 1a). Target cells infected with the respective retroviruses were subjected to phenotypic analyses at various times after selection in G-418-containing medium (Day 0), thereby eliminating cells in which a DNA copy of the retroviral genome was not incorporated into a chromosome of the host cell. Using this retroviral transduction method, we were able to overproduce both HRASV12 and cyclin E1 in human osteosarcoma U2OS cells, as revealed by western blot analysis of total cell extracts (Supplementary Fig. 1b). To assess replication fork dynamics in these cells, we used the well-established DNA fiber assay based on pulse labeling of cells with the halogenated thymidine analogues 5-chloro-2′-deoxyuridine (CldU) and 5-iodo-2′-deoxyuridine (IdU), followed by visualization of labeled replication tracts on DNA fiber spreads by indirect immunofluorescence. As expected, by measuring the length of the labeled nascent DNA tracts, we found that the rate of replication fork progression in cells overexpressing either of these oncogenes was significantly lower than in cells harboring the empty vector at both 2 and 4 days after selection (Supplementary Fig. 1c). We also observed that in cells overexpressing HRASV12 or cyclin E1, labeled nascent DNA tracts of sister replication forks showed a marked asymmetry in their length (Supplementary Fig. 1d), a phenotype known as sister fork asymmetry, which indicates the occurrence of replication fork stalling events[30]. Consistent with the replication fork stalling phenotype, cells overexpressing HRASV12 or cyclin E1 also exhibited an increased frequency of anaphase bridges and micronuclei (Supplementary Fig. 1e, f), which results from segregation of incompletely replicated DNA[31]. Furthermore, using high content microscopy followed by unbiased quantitative image-based cytometry (QIBC), we found that overexpression of either oncogene induced the formation of γH2AX foci in S-phase cells identified by PCNA staining (Supplementary Fig. 2a–c). Notably, only a mild increase

in 53BP1 foci - marker of DNA double-strand breaks (DSBs) - was detected in S/G2 cells upon overexpression of HRASV12 or cyclin E1 (Supplementary Fig. 2d, e), suggesting that replication stalling induced by these oncogenes does not lead to extensive DNA breakage during S-phase. Additionally, a modest increase in 53BP1 foci was detected in G1 cells upon overexpression of either oncogene (Supplementary Fig. 2d, e), potentially reflecting unresolved under-replicated DNA from the previous cell cycle that led to DNA breakage during chromosome segregation[31].

To elucidate the role of R-loops in DNA replication stress induced by overexpression of HRASV12 or cyclin E1, we introduced the retroviruses encoding these oncogenes into U2OS T-REx cells harboring an *RNASEH1* transgene under the control of a doxycycline-regulated CMV promoter (U2OS T-REx [RNH1-GFP])[32]. RNase H1 is an endonuclease that specifically cleaves the RNA strand in RNA:DNA hybrids, thereby eliminating R-loops and preventing R-loop-mediated replication stress[27,30,33]. By DNA fiber assay, we found that RNase H1 overexpression almost completely prevented replication fork slowing and sister fork asymmetry induced by HRASV12 or cyclin E1 overexpression in U2OS T-REx [RNH1-GFP] cells both 2 and 4 days after selection (Fig. 1a–c; Supplementary Fig. 3a, b). The same effects were observed when U2OS cells transduced with HRASV12 or cyclin E1 retroviruses were treated with the RNAPII transcription initiation inhibitor triptolide (TRP) for one hour prior to and during pulse labeling with CldU and IdU (Fig. 1d, e; and Supplementary Fig. 3c, d). In addition, by analysis of DNA replication intermediates using electron microscopy, we found that RNase H1 overexpression completely suppressed replication fork reversal induced by HRASV12 or cyclin E1 overexpression in U2OS T-REx [RNH1-GFP] (Fig. 1f). Taken together, these results suggest that replication fork stalling induced by HRASV12 or cyclin E1 overexpression is mainly caused by co-transcriptional R-loops. Consistently, anaphase bridges and micronucleation induced by HRASV12 or cyclin E1 overexpression were suppressed by RNase H1 overexpression in U2OS T-REx [RNH1-GFP] cells (Fig. 1g, h; Supplementary Fig. 3e).

## Cells overexpressing HRASV12 or cyclin E1 accumulate R-loops during S and G2 phases of the cell cycle

To investigate the molecular mechanisms underlying R-loop formation induced by overexpression of HRASV12 or cyclin E1, we used a U2OS T-REx cell line conditionally expressing catalytically inactive RNase H1 fused to green fluorescent protein [RNH1(D210N)-GFP][32]. This RNase H1 mutant, which binds but does not cleave RNA:DNA hybrids, accumulates at R-loops formed in cells and thus serves as an in vivo R-loop reporter[27,32–34]. The oncogenes were introduced into these cells by retroviral transduction, and RNH1(D210N)-GFP expression was induced by the addition of doxycycline for the last 24 h of post-selection growth (Fig. 2a). Prior to fixation, cells were pre-extracted with a Triton X-100-containing buffer to remove unbound RNH1(D210N)-GFP protein from nuclei. Fixed cells were subjected to immunofluorescence staining for PCNA to determine the DNA replication status and DAPI staining (DNA content) to determine the cell cycle stage of the individual cells (Fig. 2a). Using QIBC, we found that overexpression of either oncogene induced the formation of nuclear foci of RNH1(D210N)-GFP in S-phase cells (PCNA-positive), but not in G1 cells (PCNA-negative/low DAPI) (Fig. 2a–c), suggesting S-phase-specific R-loop formation, likely as a result of head-on TRCs. Consistently, proximity ligation assay (PLA) followed by QIBC revealed increased colocalization between PCNA and the elongating form of RNAPII in these cells (Supplementary Fig. 4a–c), indicating the occurrence of head-on TRCs[22]. However, we cannot exclude the possibility that some of these PLA foci represent co-directional RNAPII collisions behind the replication fork[35], potentially arising from oncogene-induced fork slowing. Interestingly, we also observed a significant increase in the number of RNH1(D210N)-GFP foci in G2 cells (PCNA-

negative/high DAPI) upon overexpression of either oncogene (Fig. 2c). These foci likely reflect R-loops persisting from S-phase because neither oncogene induced TRCs in G2 cells, as revealed by QIBC analysis of PCNA/RNAPII PLA foci (Supplementary Fig. 4c).

To confirm that the overexpression of HRASV12 and cyclin E1 via retroviral transduction induced R-loop formation, we isolated genomic DNA from U2OS cells transduced with the respective retroviral vector and performed slot blot analysis using the S9.6 antibody, which recognizes RNA:DNA hybrids. As expected, we observed a significantly higher R-loop signal in samples from oncogene-expressing cells compared to those of cells harboring the empty vector (Supplementary Fig. 5). Treating the genomic DNA samples with RNase H before slot blot analysis diminished the S9.6 signals, confirming the specificity of the R-loop signals detected with the S9.6 antibody (Supplementary Fig. 5).

It has been proposed that R-loops induced by oncogenic RAS result from increased transcriptional activity due to upregulation of the transcription factor TBP[9]. Consistently, we found by 5-ethynyl uridine (5-EU) pulse labeling that overexpression of HRASV12, but not cyclin E1, increased the transcriptional activity in U2OS cells (Supplementary Fig. 6a). However, QIBC analysis showed that HRASV12 overexpression increased transcription not only in S-phase cells (PCNA-positive) but also in G1 cells (PCNA-negative/low DAPI) (Supplementary Fig. 6b–d), suggesting that the observed S-phase-specific accumulation of R-loops in HRASV12-expressing cells is not solely caused by increased transcription in these cells, but may be specifically associated with TRCs.

## Suppression of replication fork reversal induces unrestrained DNA synthesis and prevents chromosome mis-segregation in cells overexpressing HRASV12 or cyclin E1

Replication fork stalling at co-transcriptional R-loops can be rescued by PARP inhibition or ZRANB3 depletion[14,27], which suppresses replication fork reversal, leading to unrestrained DNA synthesis[14,27,28,36]. Since we found that fork reversal induced by HRASV12 or cyclin E1 overexpression is caused by R-loops, we sought to test whether PARP inhibition with olaparib or siRNA-mediated depletion of ZRANB3 induces unrestrained replication fork progression in U2OS cells transduced with HRASV12 or cyclin E1 retroviral vectors. By DNA fiber assay, we found that the replication fork slowing and sister fork asymmetry phenotypes induced by these oncogenes were largely suppressed when cells were pretreated with olaparib or depleted of ZRANB3 (Fig. 3a–d; and Supplementary Fig. 7a, b). Overexpression of HRASV12 or cyclin E1 mediated by the retroviral vectors also did not induce replication fork slowing in ZRANB3 knockout U2OS cells (Fig. 3e). Moreover, the fork slowing and sister fork asymmetry phenotypes of U2OS cells overexpressing HRASV12 or cyclin E1 were rescued by depletion of the SMARCAL1 translocase (Supplementary Fig. 7c, d), which acts in conjunction with ZRANB3 to promote fork reversal[25]. To confirm these findings, we depleted ZRANB3 in hTert-immortalized human BJ fibroblasts expressing tamoxifen-inducible HRASV12 (BJ-hTert HRASV12ER-TAM) and in hTert-immortalized human retinal pigment epithelial-1 (RPE1) cells engineered to overexpress cyclin E1 from a doxycycline-regulated promoter[9,37]. As expected, we observed that overexpression of HRASV12 and cyclin E1 dramatically slowed replication fork progression in the respective cells transfected with control siRNA (Supplementary Fig. 7e, f). On the contrary, cells depleted of ZRANB3 exhibited unrestrained replication fork progression upon oncogene overexpression as observed in U2OS cells (Supplementary Fig. 7e, f), excluding a cell line-specific phenomenon. In addition, we found that PARP inhibition rescued fork slowing induced by HRASV12 or cyclin E1 overexpression in primary BJ fibroblasts (Supplementary Fig. 7g, h), excluding the possibility that this phenomenon is only seen in immortalized cells. These results prove that replication fork progression in cells overexpressing HRASV12 or cyclin E1 is restrained by fork reversal.

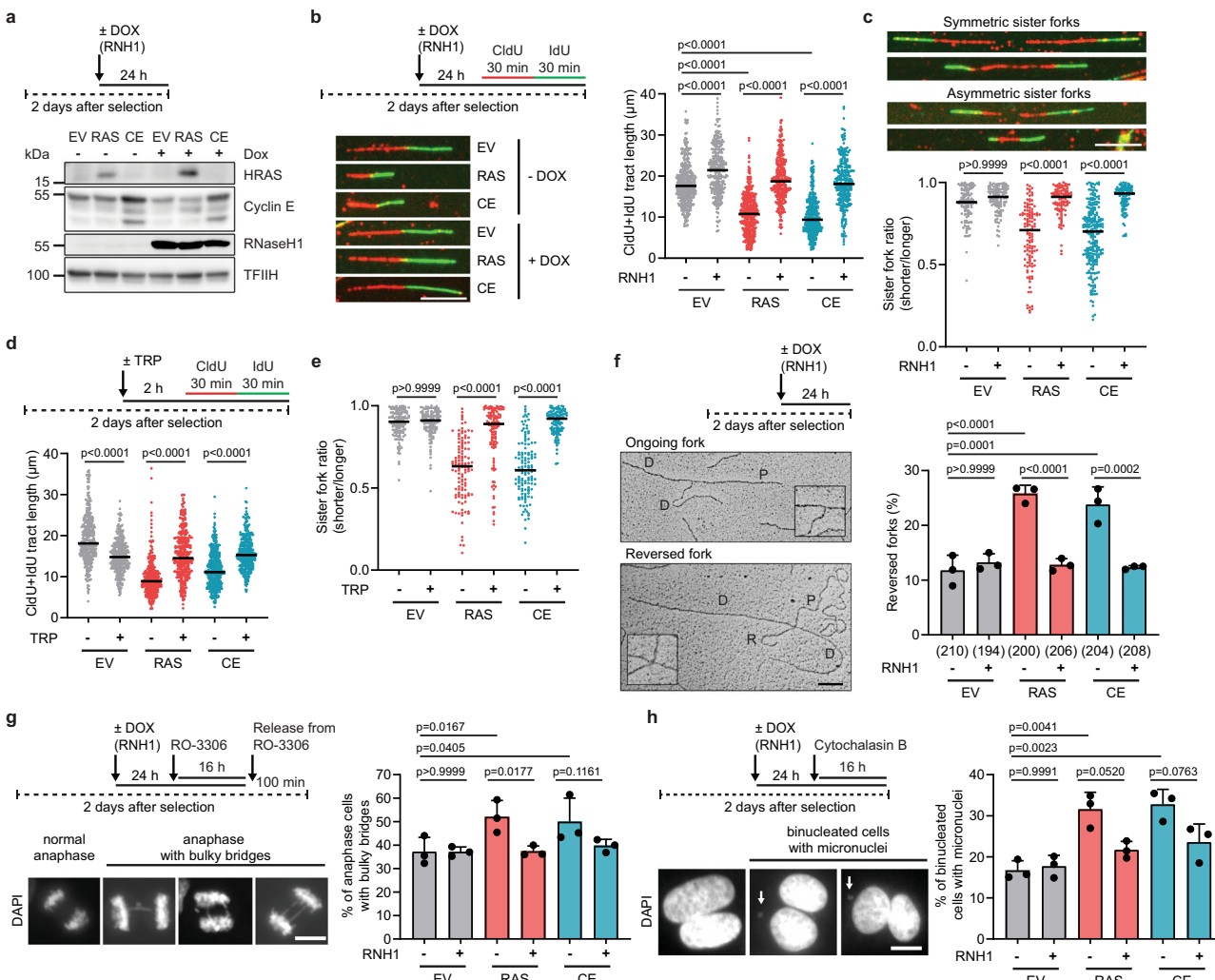

**Fig. 1 | HRASV12- and cyclin E1-induced replication stress is mainly caused by R-loops. a** Western blot analysis of HRASV12 (RAS), cyclin E1 (CE) and RNaseH1-GFP expression in transduced U2OS T-REx [RNH1-GFP] cells two days after selection. Doxycycline (DOX; 1 ng/ml) was added for the last 24 h to induce RNaseH1-GFP expression (RNH1). EV, empty vector. **b** *Top-left*: Workflow for DNA fiber labeling in transduced U2OS T-REx [RNH1-GFP] cells. *Bottom-left*: Representative images of replication tracts. *Right*: Plot of values of replication tract lengths (CldU+IdU) from three independent experiments (n ≥ 290). **c** *Top*: Representative images of symmetric and asymmetric replication tracts of sister forks for DNA fibers in (**b**). *Bottom*: Plot of values of IdU tract length ratio of sister forks (sister fork ratio) (n ≥ 102). **d** *Top*: Workflow for DNA fiber labeling in transduced U2OS cells. Transcription inhibitor triptolide (TRP, 1 μM) was present for 1 h prior to and during the labeling. *Bottom*: Plot of values of replication tract lengths (CldU+IdU) from three independent experiments (n ≥ 311). **e** Sister fork ratio for (**d**) (n ≥ 115). **f** *Top-left*: Workflow for electron microscopy (EM) analysis of replication intermediates in transduced U2OS T-REx [RNH1-GFP] cells. *Bottom-left*: Representative images of ongoing and reversed forks. Scale bar, 300 nm. P, parental arm; D, daughter arm; R,

regressed arm. The insets show magnified fork junctions: ongoing replication fork (top) and reversed fork four-way junction (bottom). *Right*: Quantification of reversed forks; the numbers of analyzed molecules are indicated in brackets. **g** *Top-left*: Workflow for enrichment of anaphase cells. Transduced U2OS T-REx [RNH1-GFP] cells were treated with 9 μM CDK1 inhibitor RO-3306 for 16 h and then released for 100 min into fresh medium. *Bottom-left*: Representative images of normal and abnormal anaphases. *Right*: Quantification of anaphase cells with bulky bridges (≥25 cells per experiment). **h** *Top-left*: Workflow for enrichment of once-divided binucleated cells by inhibition of cytokinesis. Cytochalasin B (2 μg/ml) was added to transduced U2OS T-REx [RNH1-GFP] for the last 16 h. *Bottom-left*: Representative images of binucleated cells with or without micronuclei (arrow). *Right*: Quantification of binucleated cells with micronuclei (≥100 cells per experiment). **b**– **e** Black horizontal lines indicate the median; p values were calculated by the Kruskal-Wallis test followed by Dunn's multiple comparisons test. **f**–**h** Data are presented as mean ± SD, n = 3; p values were calculated by one-way ANOVA followed by Tukey's multiple comparisons test. **b**, **c**, **g**, **h** Scale bar, 10 μm. Source data are provided as a Source Data file.

Next, we investigated whether persistent reversed forks are responsible for chromosome mis-segregation in cells overexpressing HRASV12 or cyclin E1. Notably, we found that lack of ZRANB3 completely prevented HRASV12- and cyclin E1-induced formation of anaphase bridges and micronuclei in U2OS cells (Fig. 3f, g; and Supplementary Fig. 7i). The micronucleation phenotype of these cells was also rescued by SMARCAL1 depletion (Supplementary Fig. 7j). Moreover, induction of HRASV12 or cyclin E1 expression did not increase micronucleation in ZRANB3-depleted BJ fibroblasts and RPE1 cells, respectively (Supplementary Fig. 7k, l). These findings suggest that the defective chromosome segregation in cells overexpressing

HRASV12 or cyclin E1 is caused by persistent reversed forks. Consistent with this hypothesis, we found that suppression of fork reversal by ZRANB3 depletion prevented micronucleation induced in U2OS cells by low doses of hydroxyurea (HU) or aphidicolin (Supplementary Fig. 8), both of which also induce R-loop-mediated TRCs[14,24].

## Unrestrained replication fork progression in HRASV12- or cyclin E1-overexpressing cells depends on proteins involved in restarting R-loop-stalled forks

The unrestrained replication fork progression induced by PARP inhibition in cells treated with R-loop-inducing drugs is dependent on the

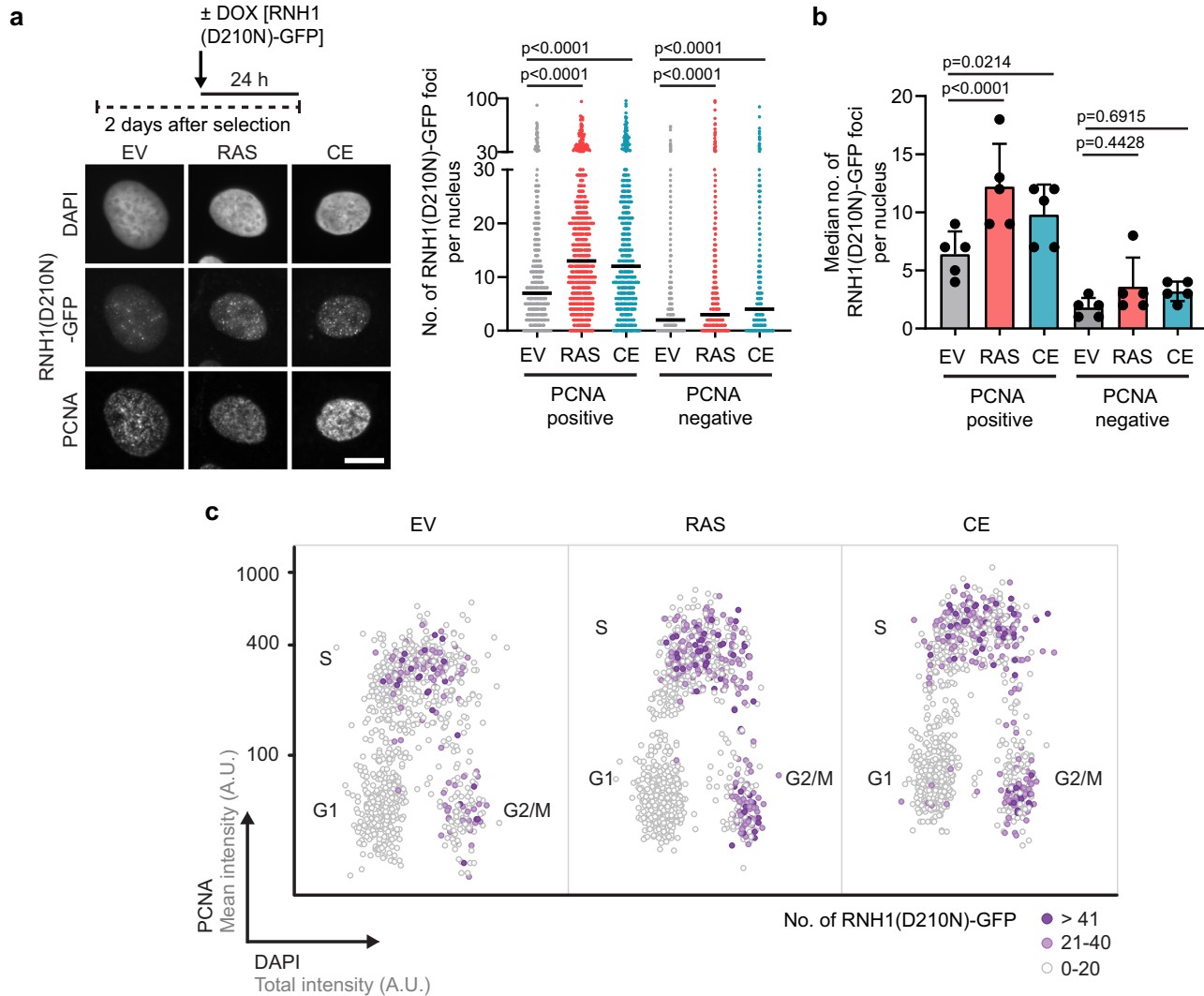

**Fig. 2 | Cells overexpressing HRASV12 or cyclin E1 accumulate R-loops during S and G2 phases of the cell cycle. a** *Top-left*: Experimental workflow. Transduced U2OS T-REx [RNH1(D210N)-GFP] cells were treated with doxycycline (DOX, 1 ng/ml) for 24 h to induce RNH1(D210N)-GFP expression. *Bottom-left:* Representative images of DAPI, RNH1(D210N)-GFP, and PCNA channels. PCNA staining was used to identify S-phase cells. Scale bar, 10 μm. *Right:* Quantification of the number of RNH1(D210N)-GFP foci per nucleus ($n \geq 463$). A representative plot from five independent experiments yielding similar results is shown. Black horizontal lines indicate the median; $p$ values were calculated by Kruskal-Wallis test followed by Dunn's multiple comparisons test. EV, empty vector; RAS, HRASV12; CE, cyclin E1. **b** Plot of the median values of the data sets represented in (**a**). Data are presented as mean ± SD, $n = 5$; $p$ values were calculated by one-way ANOVA followed by Tukey's multiple comparisons test. **c** Scatter plot of total DAPI (x-axis) and mean PCNA (y-axis) intensities in individual cells represented in (**a**). Colors indicate the number of RNH1(D210N)-GFP foci, as shown in the legend on the right. For visualization, 1000 cells per condition were randomly selected. Clusters of G1, S and G2/M phase cells are marked in the plots. A.U., arbitrary units. Source data are provided as a Source Data file.

MUS81-EME1 endonuclease, the DNA helicase RECQ1, the DNA ligase IV (LIG4), and the transcription elongation factor ELL, among others[14,27]. We therefore tested whether these factors are also required for the unrestrained DNA synthesis observed in oncogene-expressing cells upon PARP inhibition. For this, we used siRNA technology to deplete RECQ1, MUS81, LIG4 or ELL in U2OS cells transduced with the HRASV12 or cyclin E1 retroviral vectors. By DNA fiber assay, we found that depletion of any of these proteins abolished the rescue of replication fork slowing in HRASV12- or cyclin E1-expressing cells by PARP inhibition with olaparib (Fig. 4a). In addition, we found that depletion of MUS81 attenuated olaparib-stimulated DNA synthesis in HRASV12-overexpressing immortalized or primary BJ fibroblasts as well as in cyclin E1-overexpressing RPE1 cells and primary BJ fibroblasts (Supplementary Fig. 9a–d). RECQ1 promotes the restart of R-loop-stalled forks by converting reversed forks to a three-way junction, which allows for fork cleavage by MUS81 endonuclease, initiating the replication restart process[27]. Therefore, RECQ1 is not required for replication restart if fork

reversal is prevented by ZRANB3 or HLTF depletion[14,27]. Consistently, we found that unrestrained DNA synthesis induced in HRASV12- or cyclin E1-overexpressing U2OS cells by ZRANB3 depletion was dependent on the presence of MUS81 but not RECQ1 (Fig. 4b). Together, these results suggest that R-loop-mediated fork stalling in oncogene-expressing cells is counteracted by replication restart via the MUS81-LIG4-ELL axis.

## Unrestrained replication fork progression in oncogene-expressing cells defective in fork reversal is mediated by PRIMPOL

Recent work has shown that the restart of R-loop-stalled forks via the MUS81-LIG4-ELL axis also requires the DNA primase-polymerase PRIMPOL[14], which is known to mediate repriming of DNA synthesis downstream of DNA lesions that block replication fork progression[38–40]. We therefore investigated whether PRIMPOL also plays a role in the observed unrestrained replication fork progression in oncogene-expressing cells defective in fork reversal. For this, wild-

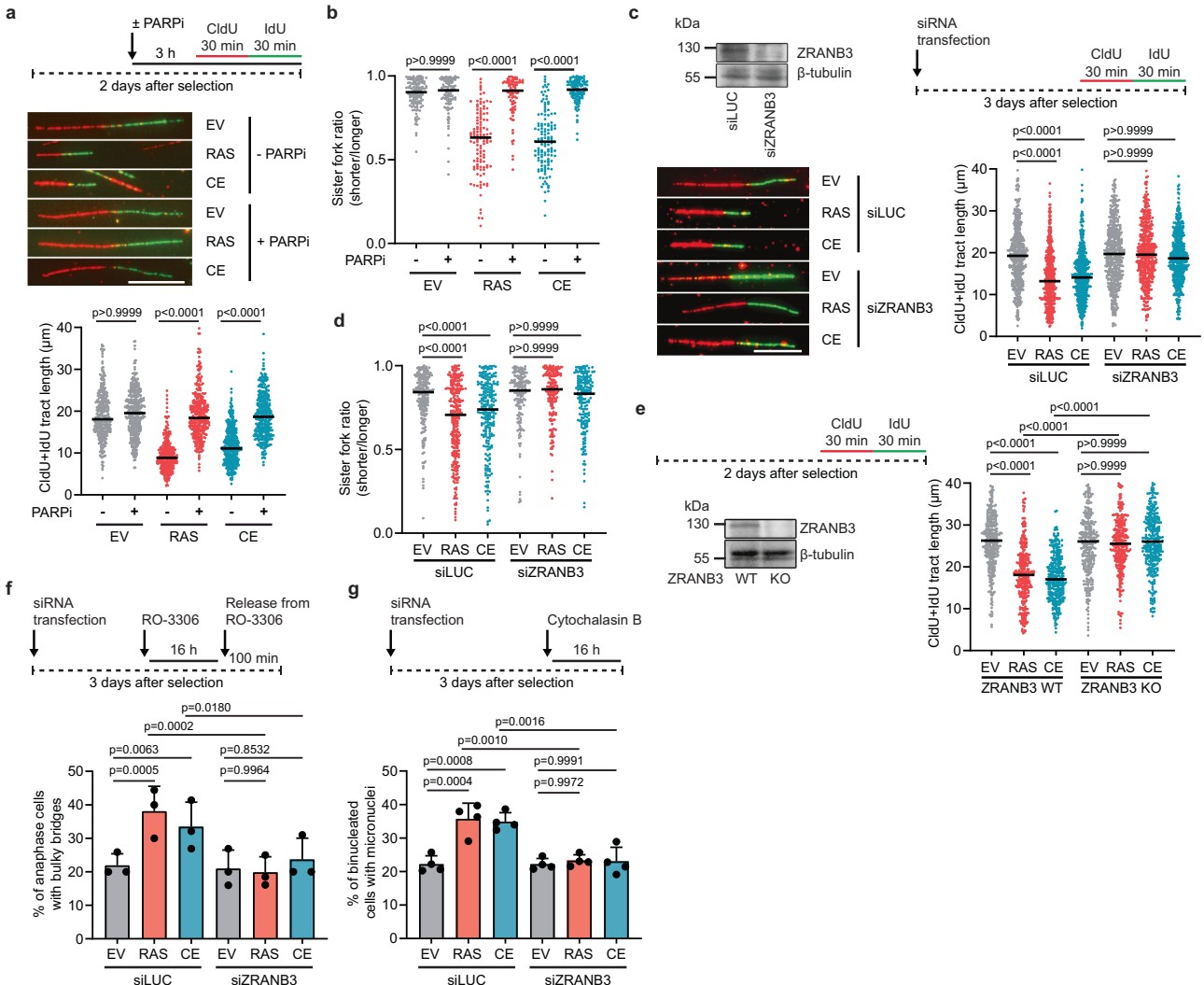

**Fig. 3 | Suppression of replication fork reversal induces unrestrained DNA synthesis and prevents chromosome mis-segregation in cells overexpressing HRASV12 or cyclin E1. a** *Top*: Workflow for DNA fiber labeling in transduced U2OS cells. PARP1 inhibitor olaparib (PARPi, 10 μM) was present for 2 h prior to and during the labeling. *Middle*: Representative images of replication tracts. Scale bar, 10 μm. EV, empty vector; RAS, HRASV12; CE, cyclin E1. *Bottom*: Plot of values of replication tract lengths (CldU+IdU) from three independent experiments ($n \geq 318$). **b** Plot of values of IdU tract length ratio of sister forks (sister fork ratio) for DNA fibers in (**a**) ($n \geq 84$). **c** *Top-left*: Western blot analysis of ZRANB3 expression in U2OS cells transfected with siLUC or siZRANB3. *Top-right*: Workflow for DNA fiber labeling in transduced U2OS cells transfected with siLUC or siZRANB3. *Bottom left*: Representative images of replication tracts. Scale bar, 10 μm. *Bottom-right*: Plot of values of replication tract lengths (CldU+IdU) from four independent experiments ($n \geq 440$). **d** Plot of values of sister fork ratio for DNA fibers in (**c**) ($n \geq 163$). **e** *Top-left*: Workflow for DNA fiber labeling in transduced U2OS ZRANB3-WT and U2OS

ZRANB3-KO cells. *Bottom-left*: Western blot analysis of ZRANB3 expression. *Right*: Plot of values of replication tract lengths (CldU+IdU) from three independent experiments ($n \geq 311$). **f** *Top*: Workflow for enrichment of anaphase cells. Transduced U2OS cells, transfected with siLUC or siZRANB3, were treated with 9 μM CDK1 inhibitor RO-3306 for 16 h, and then released for 100 min into fresh medium. *Bottom*: Quantification of anaphase cells with bulky bridges (≥25 cells per experiment). **g** *Top:* Workflow for enrichment of once-divided binucleated cells by inhibition of cytokinesis. Transduced U2OS cells, transfected with siLUC or siZRANB3, were incubated with cytochalasin B (2 μg/ml) for 16 h. *Bottom*: Quantification of binucleated cells with micronuclei (≥100 cells per experiment). **a–e** Black horizontal lines indicate the median; $p$ values were calculated by Kruskal-Wallis test followed by Dunn's multiple comparisons test. **f, g** Data are presented as mean ± SD, $n = 3$–4; $p$ values were calculated by one-way ANOVA followed by Tukey's multiple comparisons test. Source data are provided as a Source Data file.

type and ZRANB3 knockout U2OS cells transduced with HRASV12 or cyclin E1 retroviral vectors were transfected with control or PRIMPOL siRNA and replication fork progression was monitored by DNA fiber assay 72 h post-transfection. Cells were also depleted of MUS81 as a positive control. We found that, as with the proteins of the MUS81-LIG4-ELL axis, PRIMPOL depletion impaired unrestrained replication fork progression in ZRANB3 knockout cells overexpressing HRASV12 or cyclin E1 (Fig. 5a). PRIMPOL-mediated repriming causes discontinuous DNA replication, leaving a ssDNA gap behind the fork[39,40]. To confirm that this replication restart mechanism is involved in the unrestrained replication fork progression observed in HRASV12- and

cyclin E-overexpressing cells upon inactivation of fork reversal, we carried out a modified DNA fiber assay where cells were permeabilized after CldU/IdU labeling and then treated with S1 nuclease, which specifically cleaves single-stranded nucleic acids, including single-stranded regions of duplex DNA[41,42]. We found that DNA fibers in ZRANB3 knockout cells expressing HRASV12 or cyclin E1, which showed unrestrained fork progression, were sensitive to S1 nuclease digestion compared to cells carrying the empty vector (Fig. 5b). Taken together, these results indicate that unrestrained fork progression induced by fork reversal deficiency in HRASV12/cyclin E1-overexpressing cells is dependent on PRIMPOL and further support a

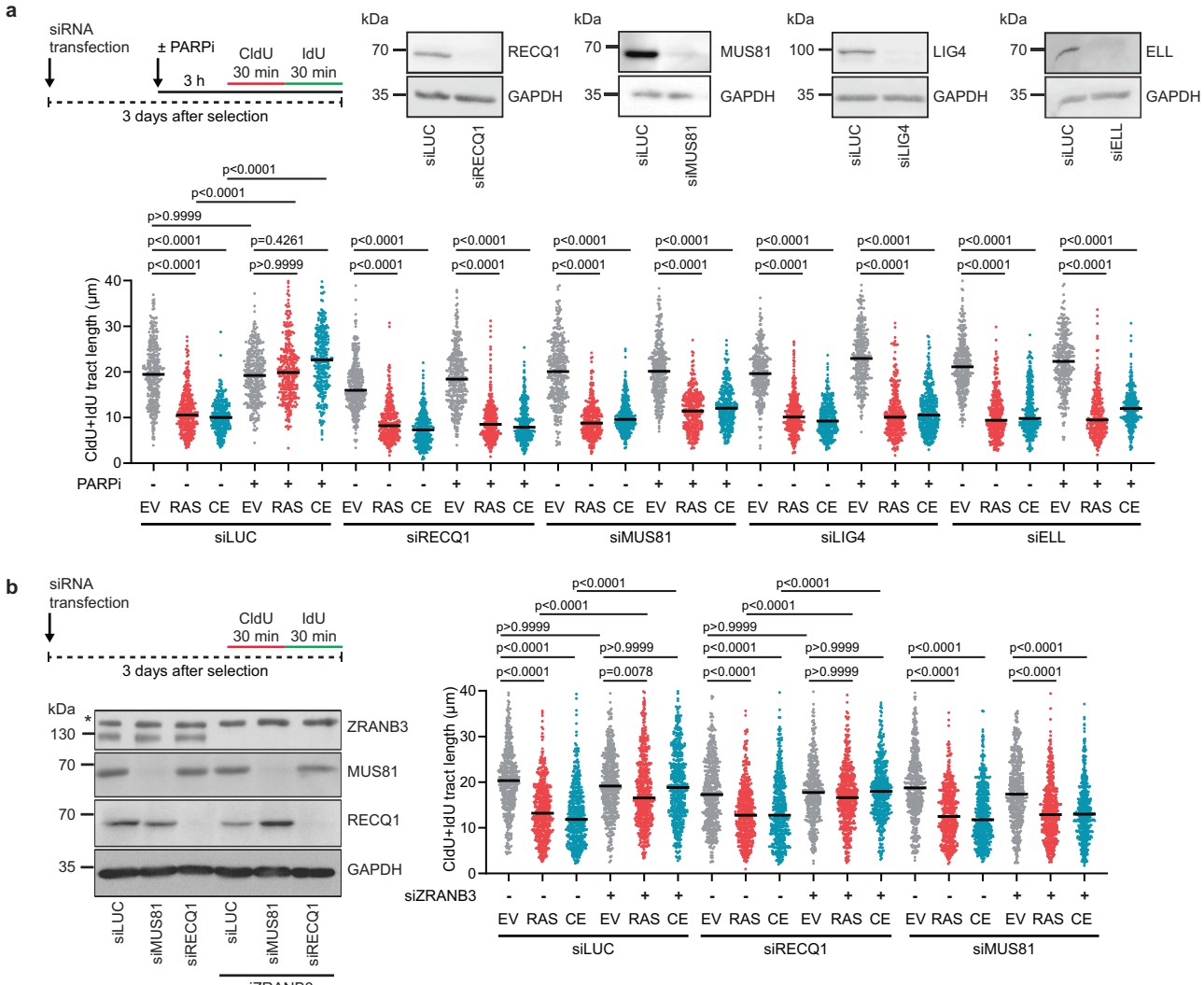

**Fig. 4 | Unrestrained replication fork progression in HRASV12- and cyclin E1-overexpressing cells depends on proteins involved in restarting R-loop-stalled forks. a** *Top-left:* Workflow for DNA fiber labeling in transduced U2OS cells 3 days after selection. Cells were transfected with appropriate siRNA on day 0. Where required, 10 μM PARP1 inhibitor olaparib (PARPi) was added 2 h before labeling and was also present during the labeling. *Top-right:* Western blot analysis of RECQ1, MUS81, LIG4 and ELL expression in U2OS cells transduced with empty vector (EV) and transfected with indicated siRNAs. *Bottom:* Plot of values of replication tract lengths (CldU+IdU) from three independent experiments (n ≥ 285).

RAS, HRASV12; CE, cyclin E1. **b** *Top-left:* Workflow for DNA fiber labeling in transduced U2OS cells 3 days after selection. Cells were transfected with appropriate siRNAs on day 0. *Bottom left:* Western blot analysis of RECQ1, MUS81 and ZRANB3 expression in U2OS cells transduced with empty vector and transfected with indicated siRNAs. The asterisk indicates non-specific band on the ZRANB3 blot. *Right:* Plot of values of replication tract lengths (CldU+IdU) from three independent experiments (n ≥ 401). **a, b** Black horizontal lines indicate the median; p values were calculated by Kruskal-Wallis test followed by Dunn's multiple comparisons test. Source data are provided as a Source Data file.

## HRASV12-induced replication stress is caused by reactive oxygen species generated by NADPH oxidases

Our finding that R-loop formation in HRASV12-overexpressing cells is caused by TRCs rather than increased transcription (Fig. 2; and Supplementary Fig. 6) prompted us to further investigate the molecular mechanism underlying HRASV12-induced replication stress. It is well established that RAS oncogenes elevate intracellular ROS levels by upregulating the NADPH oxidases NOX1 and NOX4[43-46]. In light of our recent findings that excessive ROS levels drive R-loop-mediated replication stress by promoting R-loop formation following head-on TRCs[14], we sought to determine whether HRASV12 induces replication fork

stalling through a ROS-dependent mechanism. For this, we also generated stable U2OS cell lines expressing either HRASV12 or cyclin E1 under the control of a doxycycline-inducible TRE3G promoter (Fig. 6a, b). In this system, we observed a significant reduction in replication fork speed for both oncogenes two days after the addition of doxycycline (Fig. 6a, b). To eliminate ROS, we treated cells with the ROS scavenger N-acetyl-L-cysteine (NAC), which was present 2 h prior to and during nascent DNA labeling for DNA fiber assay. Alternatively, cells were treated for 24 h with setanaxib, a dual inhibitor of NOX1 and NOX4[47,48]. We found that both NAC and setanaxib rescued the fork slowing phenotype of HRASV12-overexpressing cells but not cyclin E1-overexpressing cells (Fig. 6a, b). Similar results were obtained with U2OS cells overexpressing HRASV12 or cyclin E1 following transduction of the respective retroviral vectors (Supplementary Fig. 11a). Moreover, NAC and setanaxib rescued HRASV12-induced fork slowing in primary BJ fibroblasts (Supplementary Fig. 11b). In contrast, cyclin E1-induced fork

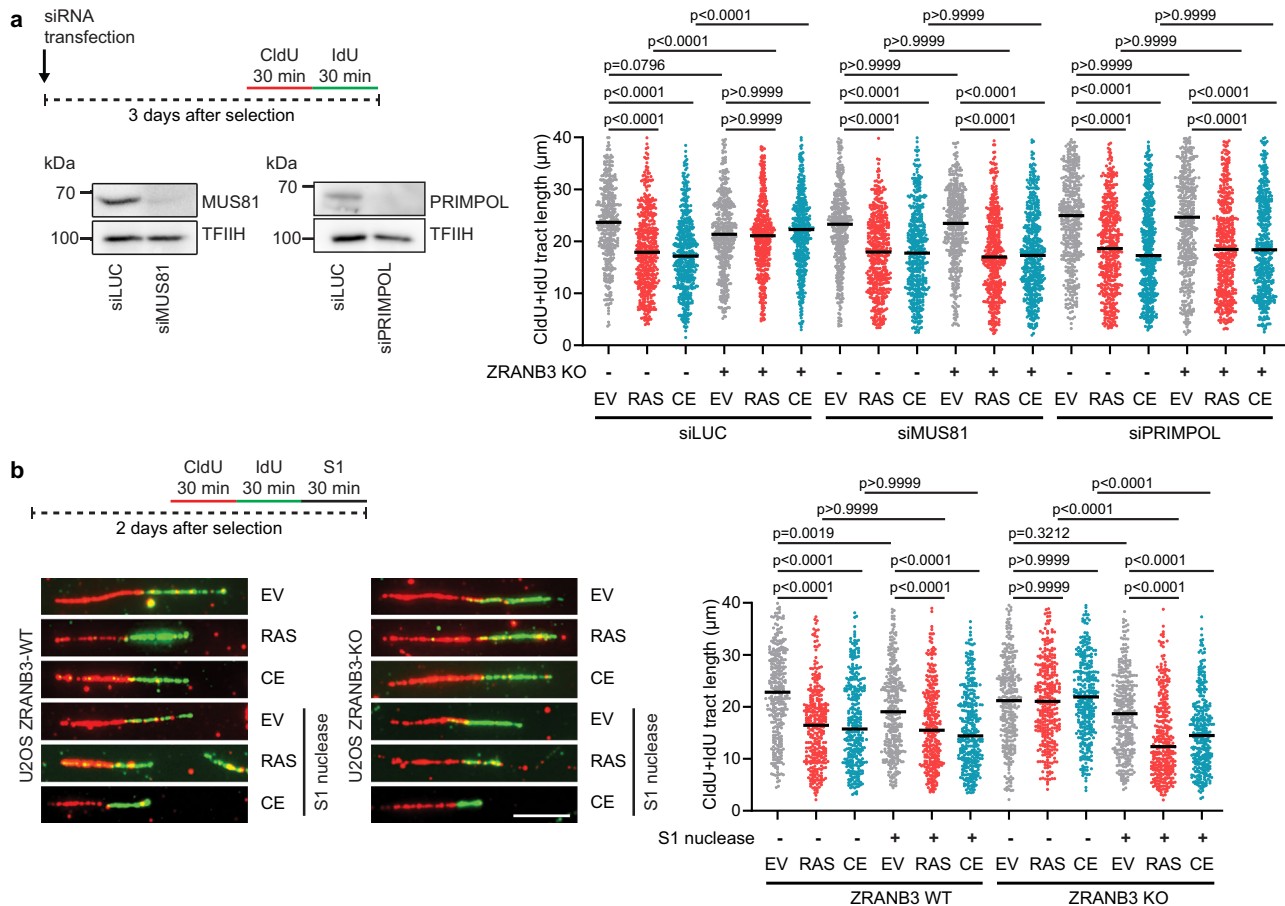

**Fig. 5 | Unrestrained replication fork progression in oncogene-expressing cells defective in fork reversal is mediated by PRIMPOL. a** *Top-left*: Workflow for DNA fiber labeling in transduced U2OS ZRANB3-WT and U2OS ZRANB3-KO cells 3 days after selection. Cells were transfected with appropriate siRNA on day 0. *Bottom-left*: Western blot analysis of PRIMPOL and MUS81 expression in U2OS ZRANB3-WT cells transfected with indicated siRNA. *Right*: Plot of values of replication tract lengths (CldU+IdU) from three independent experiments ($n \geq 433$). Cells were transduced with empty vector (EV) or vectors encoding for HRASV12 (RAS) or cyclin E1 (CE).

**b** *Top-left*: Workflow for DNA fiber/S1 nuclease assays with transduced U2OS ZRANB3-WT and U2OS ZRANB3-KO cells 2 days after selection. After CldU/IdU pulse-labeling, cells were pre-extracted and incubated with S1 nuclease to digest DNA fibers within nascent strand gaps. *Bottom-left*: Representative images of replication tracts. Scale bar, 10 μm. *Right*: Plot of values of replication tract lengths (CldU+IdU) from three independent experiments ($n \geq 316$). **a, b** Black horizontal lines indicate the median; $p$ values were calculated by Kruskal-Wallis test followed by Dunn's multiple comparisons test. Source data are provided as a Source Data file.

slowing in primary BJ fibroblasts was not affected by NAC or setanaxib, although it was still rescued by the inhibition of transcription with triptolide (Supplementary Fig. 11c). In addition, we found that HRASV12-, but not cyclin E1-induced replication fork slowing, was alleviated upon induction of antioxidant response with tert-butylhydroquinone (tBHQ) or sulforaphane (SFN) (Supplementary Fig. 11d, e), which activate NRF2 signaling[49,50]. Together, these data suggest that HRASV12-induced replication stress is caused by ROS generated by NADPH oxidase enzymes. In support of this notion, we found that micronucleation induced by HRASV12 was suppressed when cells were treated with setanaxib, tBHQ or SFN, whereas cyclin E1-induced micronucleation was not affected by these treatments (Fig. 6c, d; and Supplementary Fig. 11f, g).

## HRASV12-induced replication stress depends on the replisome-associated ROS sensor PRDX2 and is linked to PRDX2-dependent dissociation of TIMELESS from the replisome

R-loop-mediated replication fork stalling caused by excessive ROS levels depends on the replisome-associated ROS sensor PRDX2[14]. ROS disrupt the oligomeric state of PRDX2 and thereby enforce dissociation of the TIMELESS-TIPIN complex from the replisome, leading to a reduction in replication fork velocity[51]. This ultimately increases the incidence of R-loop-mediated fork stalling events

following head-on TRCs[14]. To further explore the role of ROS in HRASV12-induced replication stress, we tested whether this phenomenon is suppressed by PRDX2 depletion. By DNA fiber assay, we found that siRNA-mediated depletion of PRDX2 almost completely rescued HRASV12-induced fork slowing in U2OS cells or primary BJ fibroblasts (Fig. 7a; Supplementary Fig. 12a). PRDX2 depletion also suppressed accumulation of R-loops and micronucleus formation in HRAS-overexpressing cells (Fig. 7b–e; and Supplementary Fig. 12b, c), as did ZRANB3 depletion (Fig. 3g; and Supplementary Fig. 12d–f), which promotes replication restart[14]. Importantly, PRDX2 depletion had no effect on the replication stress phenotypes of cyclin E1-overexpressing cells (Supplementary Fig. 13a–d). These data suggest that oncogenic RAS induces R-loop-mediated fork stalling in a PRDX2-dependent manner.

To test whether HRASV12 overexpression triggers PRDX2-mediated dissociation of the TIMELESS-TIPIN complex from the replisome, we used SIRF, a PLA-based assay that enables quantitative analysis of protein interactions at active or stalled replication forks[52]. In this assay, 5-ethynyl-2′-deoxyuridine (EdU) is first incorporated into nascent DNA at replication forks, followed by formaldehyde cross-linking and EdU biotinylation via click chemistry. PLA is then performed between biotin-labeled nascent DNA and the protein of interest. We performed SIRF against TIMELESS prior to and after

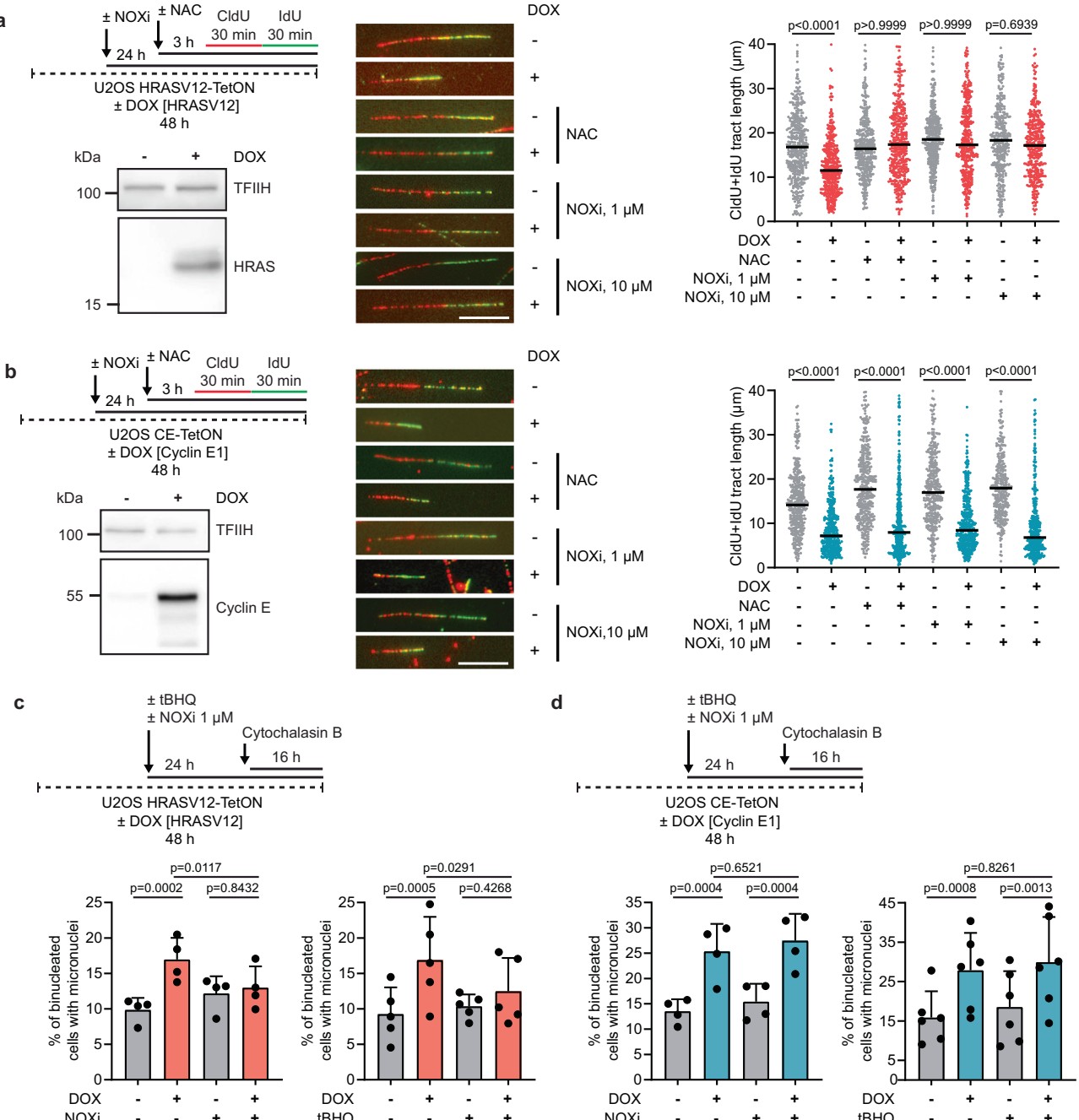

**Fig. 6 | HRASV12-induced replication stress is caused by reactive oxygen species generated by NADPH oxidases. a** *Top-left*: Workflow for DNA fiber assays with U2OS HRASV12-TetON cells. Cells were treated with doxycycline (DOX, 1 μg/ml) for 48 h to induce HRASV12 overexpression. Setanaxib - NADPH oxidase 1/4 inhibitor (NOXi, 1 μM or 10 μM), was present for 24 h prior to and during the labeling. N-Acetyl-L-cysteine (NAC, 5 mM) was added 2 h before the labeling and was also present during the labeling. *Bottom-left*: Western blot analysis of HRASV12 expression. *Middle:* Representative images of replication tracts. *Right:* Plot of values of replication tract lengths (CldU+IdU) from two independent experiments (n ≥ 367). **b** *Top-left*: Workflow for DNA fiber assays with U2OS CE-TetON cells. Cells were treated with DOX (1 μg/ml) for 48 h to induce cyclin E1 overexpression. NOXi and NAC were added as in (**a**). *Bottom-left*: Western blot analysis of cyclin E1 expression. *Middle:* Representative images of replication tracts. *Right:* Plot of values of replication tract lengths (CldU+IdU) from two independent experiments (n ≥ 340). **c** *Top:* Workflow for micronucleus assays with U2OS HRASV12-TetON cells. HRASV12 expression was induced as in (**a**). Cells were treated with tert-

butylhydroquinone (tBHQ, 10 μM) or NOXi (1 μM) for 24 h. Cytochalasin B (2 μg/ml) was added for the last 16 h to enrich for once-divided binucleated cells. *Bottom-left*: Quantification of binucleated cells with micronuclei for NOXi treatment. *Bottom-right*: Quantification of binucleated cells with micronuclei for tBHQ treatment. **d** *Top:* Workflow for micronucleus assays with U2OS CE-TetON cells. Cyclin E1 overexpression was induced as in (**b**). Cells were treated with tBHQ, NOXi and cytochalasin B as in (**c**). *Bottom-left:* Quantification of binucleated cells with micronuclei for NOXi treatment. *Bottom-right:* Quantification of binucleated cells with micronuclei for tBHQ treatment. **a, b** Black horizontal lines indicate medians; *p* values were calculated by Kruskal-Wallis test followed by Dunn's multiple comparisons test. Scale bar, 10 μm. **c, d** Data are presented as mean ± SD, *n* = 4–6; *p* values were calculated by one-way ANOVA followed by Tukey's multiple comparisons test. For each condition, at least 100 binucleated cells were examined for the presence of micronuclei in each experiment. Source data are provided as a Source Data file.

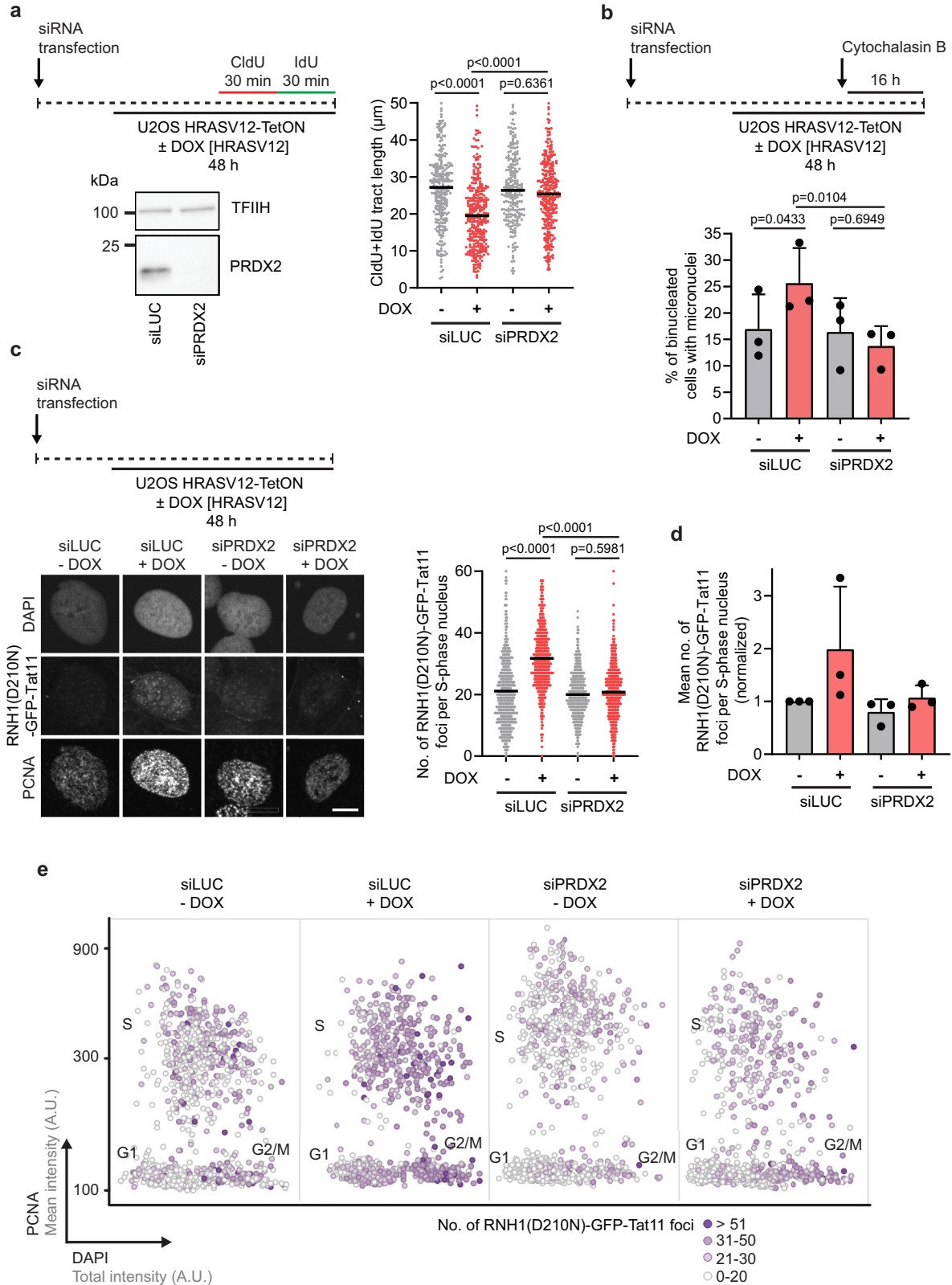

induction of HRASV12 expression in U2OS-HRASV12-TetON cells transfected with control or PRDX2 siRNA. Concomitantly, we co-clicked EdU with Alexa Fluor 647-azide to mark S-phase cells. QIBC analysis revealed that the TIMELESS-SIRF signal in S-phase nuclei decreased significantly upon HRASV12 overexpression in control siRNA-transfected cells. This reduction was not observed in PRDX2-depleted cells, suggesting that HRASV12 induces PRDX2-dependent

dissociation of TIMELESS from the replisome (Fig. 8a, b). In contrast to HRASV12, overexpression of cyclin E1 only mildly reduced the association of TIMELESS with the sites of DNA replication in PRDX2-proficient cells, with a further decrease observed upon PRDX2 depletion (Supplementary Fig. 13e, f). These findings support the conclusion that R-loop-mediated fork stalling induced by oncogenic RAS is a consequence of ROS-induced dissociation of the TIMELESS-TIPIN

**Fig. 7 | HRASV12-induced replication stress depends on the replisome-associated ROS sensor PRDX2. a** *Top-left*: Workflow for DNA fiber assays with U2OS HRASV12-TetON cells transfected with siLUC or siPRDX2. Cells were treated with doxycycline (DOX, 1 µg/ml) for 48 h to induce HRASV12 overexpression. *Bottom-left*: Western blot analysis of PRDX2 expression. *Right*: Plot of values of replication tract length (CldU+IdU) from two independent experiments (n ≥ 236). Black horizontal lines indicate the median; *p* values were calculated by Kruskal-Wallis test followed by Dunn's multiple comparisons test. **b** *Top*: Workflow for micronucleus assays with U2OS HRASV12-TetON transfected with siLUC or siPRDX2. HRASV12 expression was induced as in (**a**). Cytochalasin B (2 µg/ml) was added for the last 16 h to enrich for once-divided binucleated cells. *Bottom*: Quantification of binucleated cells with micronuclei. Data are presented as mean ± SD, *n* = 3; *p* values were calculated by one-way ANOVA followed by Tukey's multiple comparisons test. For each condition, at least 100 binucleated cells were examined for the presence of micronuclei in each experiment. **c** *Top-left*: Workflow for R-loop assays with U2OS HRASV12-TetON cells transfected with siLUC or

siPRDX2. Cells were treated with DOX (1 µg/ml) for 48 h to induce HRASV12 overexpression. Cells were stained with recombinant RNH1(D210N)-GFP-Tat11 protein and PCNA antibody. *Bottom-left:* Representative images of DAPI, RNH1(D210N)-GFP-Tat11, and PCNA channels. Scale bar, 10 µm. *Right:* Quantification of the number of RNH1(D210N)-GFP-Tat11 foci per S-phase nucleus (*n* = 623). A representative plot from three independent experiments yielding similar results is shown. Black horizontal lines indicate the mean; *p* values were calculated by Kruskal-Wallis test followed by Dunn's multiple comparisons test. **d** Plot of the mean values of the data sets represented in (**c**). Data are normalized to the siLUC/DOX(-) condition. Error bars represent SD, *n* = 3. **e** Scatter plot of total DAPI (x-axis) and mean PCNA (y-axis) intensities in individual cells represented in (**c**). Colors indicate the number of RNH1(D210N)-GFP-Tat11 foci, as shown in the legend on the right. For visualization, 1000 cells per condition were randomly selected. Clusters of G1, S and G2/M phase cells are marked in the plots. A.U., arbitrary units. Source data are provided as a Source Data file.

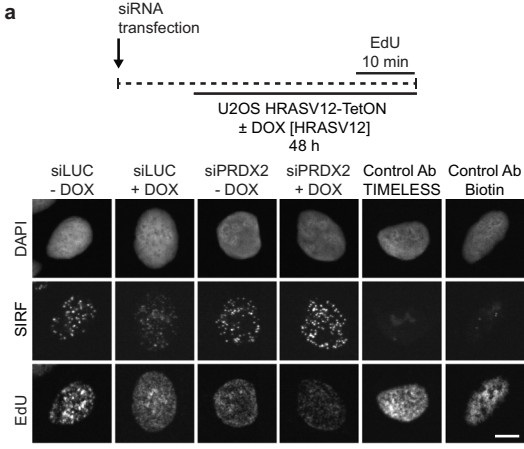
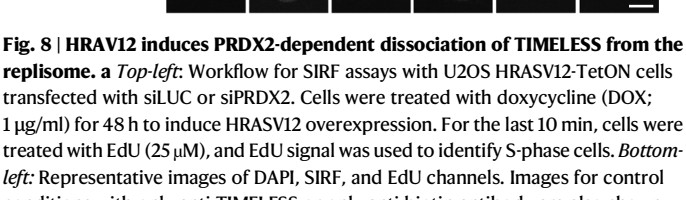
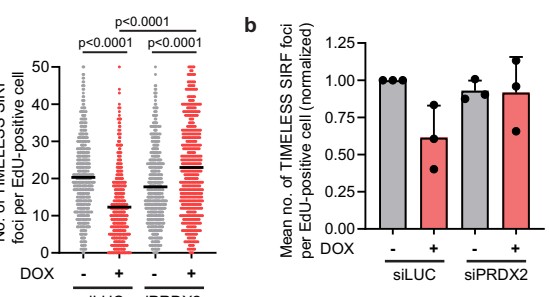

**Fig. 8 | HRAV12 induces PRDX2-dependent dissociation of TIMELESS from the replisome. a** *Top-left*: Workflow for SIRF assays with U2OS HRASV12-TetON cells transfected with siLUC or siPRDX2. Cells were treated with doxycycline (DOX; 1 µg/ml) for 48 h to induce HRASV12 overexpression. For the last 10 min, cells were treated with EdU (25 µM), and EdU signal was used to identify S-phase cells. *Bottom-left:* Representative images of DAPI, SIRF, and EdU channels. Images for control conditions with only anti-TIMELESS or only anti-biotin antibody are also shown.

Scale bar, 10 µm. *Right:* Quantification of the number of TIMELESS SIRF foci per S-phase nucleus (*n* ≥ 543). A representative plot from three independent experiments yielding similar results is shown. Black horizontal lines indicate the mean; *p* values were calculated by Kruskal-Wallis test followed by Dunn's multiple comparisons test. **b** Plot of the mean values of the data sets represented in (**a**). Data are normalized to the siLUC/DOX(-) condition. Error bars represent SD, *n* = 3. Source data are provided as a Source Data file.

complex from the replisome, which slows down replication fork progression.

## Discussion

In this study, we demonstrate that replication fork reversal and chromosome mis-segregation induced by HRASV12 or cyclin E1 overexpression are mainly caused by R-loops that appear to form specifically in S-phase, possibly during transcription-replication encounters. Evidence suggests that head-on TRCs induce R-loop formation and subsequent fork reversal when replication fork progression is slowed[14]. This can occur, for instance, through partial inhibition of replicative DNA polymerases with aphidicolin or under conditions of excessive generation of ROS[14], which promote dissociation of the fork acceleration factor TIMELESS-TIPIN from the replisome in a manner dependent on the replisome-associated ROS-sensor PRDX2[51]. Interestingly, HRASV12 has been shown to increase cellular ROS levels by upregulating the NADPH oxidase NOX4[44]. Moreover, HRASV12-induced DNA damage and subsequent senescence can be suppressed by NOX4 depletion[44]. Here, we show that HRASV12-induced replication stress is alleviated by ROS scavenging, NOX1/4 inhibition, or PRDX2 depletion, whereas cyclin E1-induced replication stress is unaffected by these treatments. Furthermore, we demonstrate that HRASV12 induces TIMELESS dissociation from replication forks in a manner dependent

on PRDX2. Thus, the R-loop-mediated replication fork stalling observed upon HRASV12 expression may be driven by ROS-induced dissociation of the TIMELESS-TIPIN complex from the replisome, resulting in replication slowdown (Fig. 9). In support of this hypothesis, overexpression of TIMELESS has been shown to rescue HRASV12-induced replication stress[53]. We propose that the slow progression of replication forks towards head-on transcription complexes leads to a gradual buildup of positive DNA supercoiling in the DNA template, which impairs transcription elongation and promotes R-loop formation; these R-loops, in turn, induce replication fork reversal and thereby halt fork progression (Fig. 9).

It should be noted that HRASV12-induced replication stress has been shown to depend on the TATA-box binding protein TBP, whose expression is induced by RAS signaling[9]. Moreover, overexpression of TBP itself was found to cause replication stress and genomic instability[9]. Based on these findings, it has been proposed that replication stress induced by HRASV12 arises from increased global transcription driven by TBP upregulation, which promotes R-loop formation[9]. However, based on our findings, it is more likely that the reported rescue of HRASV12-induced replication stress by TBP depletion results from downregulation of the NOX1/4 enzymes, which would be expected in the case of a deficiency in a general transcription factor such as TBP. Nevertheless, we cannot exclude the possibility that

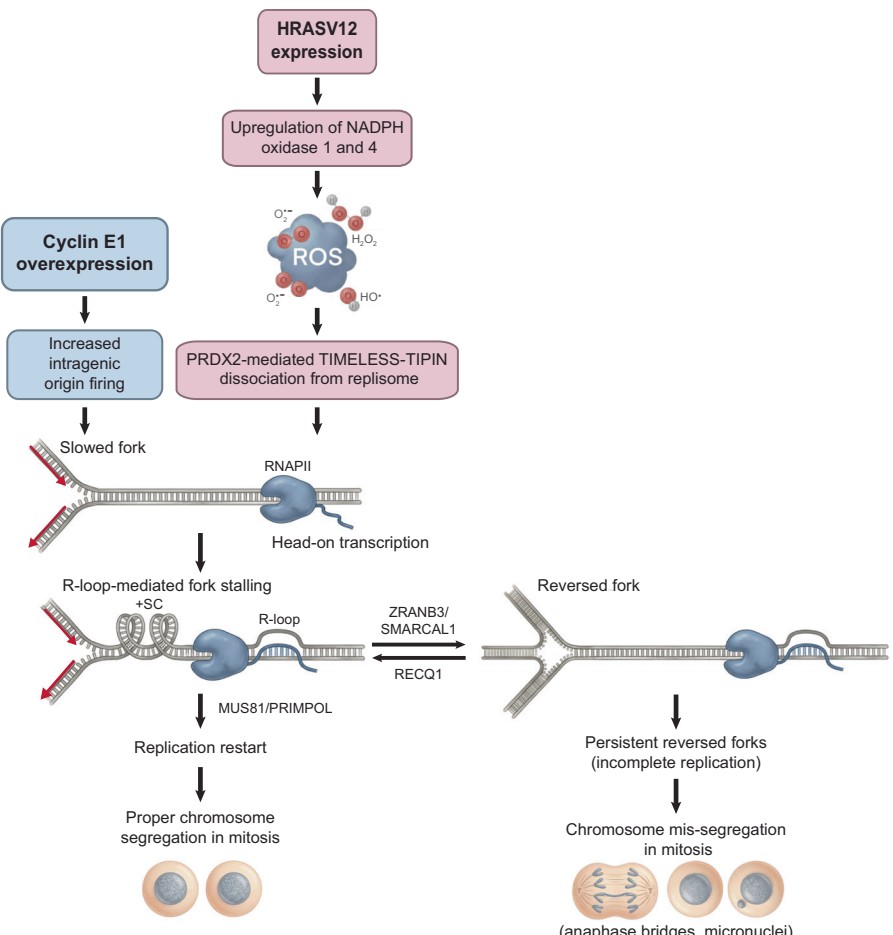

**Fig. 9 | Model for the molecular mechanisms involved in HRASV12- and cyclin E1-induced replication stress and its resolution.** DNA replication stress induced by HRASV12 oncogene is driven by excessive reactive oxygen species (ROS) generated by the upregulation NADPH oxidase 1 and 4. This leads to PRDX2-mediated dissociation of the TIMELESS-TIPIN complex from the replisome, resulting in replication fork slowdown. The slow progression of replication forks towards head-on transcription complexes causes a gradual buildup of positive DNA supercoiling (+SC) in the DNA template, which promotes R-loop formation followed by ZRANB3/SMARCAL1-mediated fork reversal. Cyclin E1 overexpression increases intragenic origin firing, enhancing the frequency of head-on transcription-replication conflicts and concomitantly reducing replication fork velocity due to excessive origin firing, ultimately leading to the same outcome as observed with HRASV12. R-loop-mediated fork reversal is counteracted by RECQ1 DNA helicase, which unwinds reversed forks and promotes replication restart via the MUS81-LIG4-ELL pathway involving PRIMPOL repriming. This ensures the completion of DNA replication and proper chromosome segregation in mitosis. Persistent reversed forks impair chromosome segregation, giving rise to anaphase bridges and micronuclei, which can lead to chromosomal rearrangements. RNAPII, RNA polymerase II.

increased transcription activity enhances R-loop formation at sites of head-on TRCs in HRASV12-expressing cells. Interestingly, several other oncogenes, including MYC, BCR/ABL, and BRAF, are known to elevate cellular ROS levels[4]. It will therefore be informative to determine whether these oncogenes also induce R-loop-mediated replication stress in a ROS- and PRDX2-dependent manner.

DNA replication stress induced by overexpression of cyclin E1 results from aberrant firing of intragenic replication origins, which occurs due to premature entry of cells into S-phase and leads to head-on TRCs[19]. Interestingly, increased origin activation, a condition observed in cyclin E1-overexpressing cells, has been shown to reduce replication fork velocity, possibly due to depletion of factors that are essential for proper replication fork progression[54]. Thus, it is plausible that the aforementioned mechanism, whereby conflicts between slowed replisomes and head-on transcription complexes lead to R-loop formation and subsequent fork reversal, also underlies cyclin E1-induced replication stress (Fig. 9). This hypothesis is supported by the observation that downregulation of replication initiation by CDC7 inhibition rescues replication fork slowing in cyclin E1-overexpressing cells[18]. Notably, we found that CDC7 inhibition also rescued HRASV12-

induced fork stalling, possibly by increasing fork velocity and thereby preventing R-loop formation at TRC sites (Supplementary Fig. 14). Thus, our findings demonstrate how different oncogenic triggers converge on a common outcome: R-loop-mediated TRCs, a key driver of DNA replication stress and genomic instability (Fig. 9).

In this study, we found that inhibition of fork reversal induces unrestrained fork progression and prevents chromosome mis-segregation in cells overexpressing HRASV12 or cyclin E1 (Fig. 3). Moreover, fork progression under these conditions was found to be dependent on the proteins of the MUS81-LIG4-ELL axis implicated in restarting R-loop-stalled forks (Fig. 4). These findings provide further support for our proposal that replication stress induced by oncogenes such as HRASV12 or cyclin E1 is caused by R-loop-mediated TRCs. Furthermore, our findings suggest that chromosomal instability induced by these oncogenes is caused by persistent reversed forks, which prevent the completion of DNA replication, thereby impairing proper chromosome segregation during mitosis. Of note, fork reversal was also found to account for micronucleation induced by low doses of aphidicolin or HU (Supplementary Fig. 8), which also induce R-loop-mediated TRCs[14]. Thus, despite preventing fork breakage during S

phase, fork reversal can be a source of chromosomal instability if the reversed forks are not restarted before the onset of anaphase (Fig. 9).

We found that unrestrained DNA synthesis induced in oncogene-overexpressing cells upon suppression of fork reversal was also dependent on PRIMPOL. PRIMPOL, in addition to the proteins of the MUS81-LIG4-ELL axis, was also required for unrestrained fork progression induced by fork reversal deficiency in cells exposed to cisplatin or low concentrations of HU[14,41,42]. Moreover, here, we show that PRIMPOL acts in a common pathway with MUS81 to prevent accumulation of stalled forks (Supplementary Fig. 10). Collectively, these data strongly suggest that repriming by PRIMPOL is part of the replication restart mechanism mediated by the MUS81-LIG4-ELL axis at R-loop-stalled forks.

Interestingly, a recent study has shown that DNA replication stress induced by KRAS oncogene was suppressed by overexpression of the ATR kinase in a manner dependent on PRIMPOL[11]. Elevated ATR expression also suppressed the formation of micronuclei and promoted cell survival upon KRAS[G12V] expression[11]. Of note, ATR is known to phosphorylate SMARCAL1 to prevent its fork reversal activity[55]. It is therefore plausible that this PRIMPOL-dependent unrestrained fork progression induced by ATR in KRAS[G12V]-expressing cells is driven by a fork reversal deficiency and is mediated by the MUS81-LIG4-ELL axis. These findings also provide further support for our proposal that MUS81/PRIMPOL-dependent DNA synthesis counteracts R-loop-induced replication fork reversal to prevent chromosome damage in mitosis upon oncogene activation.

Genetic and epigenetic alterations in cancer cells may confer specific tumor vulnerabilities that can be targeted therapeutically[56]. In this work, we have established that inhibition of replication fork reversal rescues oncogene-induced replication stress in a manner dependent on the proteins involved in the restart of R-loop-stalled replication forks, including MUS81 endonuclease. This finding suggests that disrupting pathways responsible for TRC resolution might compromise cell survival upon induction of R-loop-dependent replication stress. Interestingly, silencing of the *HLTF* gene through promoter methylation is frequently observed in human colon cancers, but not in normal colon tissue[57]. Colon cancers also frequently carry oncogenic *KRAS* mutations[7]. Given that HLTF is essential for replication fork reversal[41], it is tempting to speculate that HLTF-deficient cancers rely on MUS81-mediated fork restart for survival, particularly when expressing oncogenic RAS. In support of this hypothesis, we have found that MUS81 depletion confers hypersensitivity to R-loop-inducing drugs such as camptothecin or pyridostatin in human cells[27]. In addition, we have shown that MUS81-deficient cells exhibit R-loop-dependent hypersensitivity to the PARP inhibitor olaparib, which suppresses fork reversal[27]. It is important to note that small molecule inhibitors of the MUS81 endonuclease are currently being developed by AstraZeneca[58]. These inhibitors could be used to test whether MUS81 deficiency is synthetically lethal with loss of fork reversal activity and whether this effect is enhanced by oncogenic RAS.

## Method

### Plasmid constructions and protein purification
The plasmids pBABEneo-HRASV12 was kindly provided by Dr. Fabrizio d'Adda di Fagagna. For generation of empty vector (pBABEneo), pBABEneo-HRASV12 was digested with BamHI and EcoRI to excise the HRASV12 insert. The cleaved vector was religated by T4 DNA ligase (NEB, M0202S) after filling in DNA ends with Klenow Fragment (NEB, M0210S). pBABEneo-CCNE1 was constructed by inserting the Bsp1407I/HindIII fragment of pLXSNneo-CCNE1 (kindly provided by Dr. Stefano Ferrari) including the *CCNE1* gene between the Bsp1407I and HindIII sites in pBABEneo. The RNH1(D210)-GFP sequence was amplified by PCR from the plasmid pAIOhM27RNaseH1-EGFP D210N[32] using the following primers: 5′-GGATCACATATGTTCTATGC CGTGAGGAGG-3′ (forward); 5′-CATGAGAATTCCCATCTTGTACAGC

TCGTCCATGCC-3′ (reverse). The resulting PCR product was digested with NdeI and EcoRI and cloned along with the EcoRI/BspEI fragment of pQE60-yRad52-Tat11-His₆[59] in the plasmid pTXB1 (New England Biolabs) to create a bacterial expression vector for RNH1(D210N)-GFP-Tat11-His6 fusion protein. Lentiviral vectors for inducible expression of HRASV12 and cyclin E1, respectively, were constructed by PCR amplification of the respective ORF [primers for *HRASV12*: 5′-GCAGCA-GAATTCGCCACCATGACTGAATACAAG-3′ and 5′-GCAGCAGGATCCTC AGGACAGCACACATTTGC-3′; primers for *CCNE1*: 5′-GCAGCAGAATT CGCCACCATGCCGAGGGAGCGCAGGG-3′ and 5′-GCAGCAGGATCCT-CACGCCATTTCCGGCCCGC-3′] and inserting the resulting PCR product into pLVX-TetOne-Puro (Clontech Laboratories, 631849) between EcoRI and BamHI sites.

RNH1(D210N)-GFP-Tat11-His6 protein was produced in *E. coli* BL21-CodonPlus (DE3)-RIL (Agilent Technologies, 230245) transformed with the above pTXB1-base vector. Protein expression was induced by 0.1 mM β-D-1-thiogalactopyranoside (IPTG) at 16 °C for 16 h. Cells were lysed by sonication in buffer A [50 mM Tris-HCl (pH 7.5), 300 mM NaCl, 10% (v/v) glycerol, and 2 mM 2-Mercaptoethanol] supplemented with 10 mM imidazole, 0.1 mM phenylmethylsulfonyl fluoride (PMSF), and protease inhibitor cocktail (Roche cOmplete, EDTA free). Cell extract was clarified by ultracentrifugation and loaded onto 5-ml HisTrap FF column (Cytiva, 17531901). Bound proteins were eluted with a linear gradient of imidazole (40–300 mM) in buffer A. Peak fractions with RNH1(D210N)-GFP-Tat11-His6, determined by SDS-PAGE, were pooled and diluted with buffer B [50 mM Tris-HCl (pH 7.5), 0.1 mM EDTA, 10% (v/v) glycerol, and 1 mM dithiothreitol (DTT)] to a final concentration of NaCl of 150 mM. The sample was loaded onto a 5-ml HiTrap SP HP column (Cytiva, 17115101) equilibrated with buffer B/150 mM NaCl. Bound proteins were eluted with a 50-ml linear gradient of NaCl (150–500 mM) in buffer B. Peak fractions were pooled, dialyzed against 25 mM Tris-HCl (pH 7.5), 20% (v/v) glycerol, 0.1 mM EDTA 110 mM NaCl, and 0.1 mM PMSF, and stored at −80 °C.

### Cell culture
The following human cells lines we used in this study: U2OS (ATCC; HTB-96), HEK293T (ATCC; CRL-3216), BJ-hTert HRASV12ER-TAM[9], RPE1 CE-TetON (RPE1-pRetroX-Tet-On Advanced cells stably transfected with pRetroX-Tight-Pur-CCNE1)[60], U2OS ZRANB3-WT, U2OS ZRANB3-KO[36], Phoenix-AMPHO (ATCC; CRL-3213), HeLa Kyoto (Cancer Research UK, CVCL_1922), HeLa Kyoto MUS81 knockout[27], and U2OS T-REx (Invitrogen) cell lines carrying pAIO-based vectors encoding for RNaseH1-GFP or RNaseH1(D210N)-GFP[32]. Cells were cultivated in Dulbecco's modified Eagle's medium (DMEM; Gibco, 41966-029) supplemented with 10% (v/v) fetal bovine serum (FBS; Tet-free approved, Gibco, 10270), 100 U/ml penicillin and 100 µg/ml streptomycin (Thermo Fisher Scientific, 15140122). Cells stably transfected with pAIO-based plasmids were selected in the presence of hygromycin B (50 µg/ml; Sigma-Aldrich, H3274) and puromycin (1 µg/ml, Sigma-Aldrich, P8833). Doxycycline (Takara Bio, 631311) was added for 24 h to induce overexpression of the RNaseH1 variants in U2OS T-REx cells (1 ng/ml) and for 72 h to induce overexpression of cyclin E1 in RPE1 CE-TetON (1 µg/ml). 4-Hydroxytamoxifen (Sigma-Aldrich, SML1666) at a concentration of 333 nM was added for 72 h to induce overexpression of HRASV12 in BJ-hTert HRASV12ER-TAM cells. Primary BJ HRASV12-TetON and BJ CE-TetON fibroblasts were obtained from Prof. Jiri Bartek[6], and grown in DMEM, low glucose (Gibco, 31885-023) supplemented with 10% (v/v) FBS, 100 U/ml penicillin, 100 µg/ml streptomycin and non-essential amino acids (Gibco, 11140050). Oncogene expression was induced by addition of doxycycline (2 µg/ml) for 48 h.

### Retroviral transduction
For production of retroviruses, $5.5 \times 10^6$ of Phoenix-AMPHO cells were seeded to a 10-cm Petri dish 24 h prior to transfection of

retroviral vectors. 1 h before transfection, cells were treated with 25 μM chloroquine (Sigma-Aldrich, C6628). For transfection of retroviral vectors, 500 μl of 2x HBS (50 mM HEPES, 10 mM KCl, 12 mM Dextrose, 280 mM NaCl, 1.5 mM Na₂HPO × 7H₂O; pH 7.04) were mixed with 500 μl CaCl₂ plasmid DNA mix [61 μl of 2 M CaCl₂, 10 μg of retroviral vector], incubated for 5 min at room temperature (RT) and added drop-wise to the plate with Phoenix-AMPHO cells containing 5.5 ml of medium. After 8 h, the medium was replaced with fresh medium and this was repeated once again 24 h after transfection. 48 h post-transfection, the medium containing retroviral particles produced by Phoenix cells was collected, sterile-filtered (0.22 μm, Millipore) and polybrene (Sigma-Aldrich, H9268-5G) was added to a concentration of 8 μg/ml. This virus-containing medium was then added to target cells, which were seeded 24 h before the transduction at a density of $1.2 \times 10^6$ cells per 10-cm Petri dish. After a 3-h incubation, the medium was changed to freshly filtered virus-containing medium, and the plate was again incubated for 3 h. After this second transduction step, target cells were cultured in non-viral medium overnight. The next day, third retroviral transduction was done. After 3 h, cells were incubated for 18 h in a fresh non-viral medium. The following day, transduced and non-transduced cells were re-plated to 50% confluency in growth medium containing the selection antibiotic G-418 (500 μg/ml; Thermo Fisher Scientific, 10131027). The selection took typically 96 h, when the non-transduced control died (Day 0). The transduced cells were then used for experiments. Phenotypic analyses were conducted 2 to 4 days after the end of selection. Where specified, cells were transfected with siRNAs on the day 0 and harvested 72 h later.

## Generation of stable cell lines *via* lentiviral transduction

For lentivirus production, HEK293T cells (Invitrogen) were transfected with the packaging plasmids pVSV, pMDL and pREV (kindly provided by Prof. Beat Schäfer), and the expression plasmid pLVX-TetOne-Puro-HRASV12 or pLVX-TetOne-Puro-CCNE1 using TransIT-X2 Dynamic Delivery System (Mirus, MIR 6004) according to the manufacturer's instructions. 16 h after the transfection, medium was replaced with fresh medium. 48 h post-transfection medium containing lentiviral particles produced by HEK293T cells was collected, filtered through a 0.45 μm filter syringe and supplemented with polybrene (10 μg/ml). This virus-containing medium was then added to target U2OS cells which were seeded 24 h before transduction. After 24 h incubation, the medium was replaced with fresh virus-free medium. Transduced cells were selected in the presence of puromycin (1 μg/ml; Sigma-Aldrich, P8833). Clones were isolated after 10 to 14 days of growth in the selection medium. The expression of HRASV12 and cyclin E1 after induction with doxycycline (1 μg/ml, 48 h) was tested by western blotting.

## Small-interfering RNA transfection

Transfections of siRNAs (a final concentration of 40 nM, in the case of double depletion 20 nM each) were done using Lipofectamine RNAiMAX (Thermo Fisher Scientific, 13778150) according to the instructions of manufacturer (reverse transfection protocol). Cells were typically seeded 6 h before siRNA transfection to 50% confluency. 24 h after siRNA transfection, the medium was replaced with fresh medium, and cells were replated if needed. Experiments were carried out 72 h after siRNA transfection. The majority of siRNA oligonucleotides used in this study were purchased from Microsynth AG. The sequences of the sense strand of siRNA duplexes are: siLUC (CGUACGCGGAAUACUUCGA); siMUS81 (CAGCCCUGGUGGAUCGAUA); siLIG4 (GCUAGAUGGUGAACGUAUG); siPRIMPOL (GAGGAAAGCUGGACAUCGA); siRECQ1 (GCAAGGAGAUUUACUCGAA); siSMARCAL1 (UUGCUAAGAAGGUCAAAGC); siPRDX2 (AGAUCAUCGCGUUCAGCAA). ZRANB3 siRNA was purchased from Dharmacon (84083, D-010025-03-0005), ELL siRNA was purchased from Santa Cruz Biotechnology (sc-38041).

## DNA fiber assay

Cells were pulse-labeled with 30 μM thymidine analog 5-chloro-2′-deoxyuridine (CldU) (Sigma-Aldrich, C6891) for 30 min, then washed three times with PBS (pre-warmed to 37 °C), followed by pulse-labeling with 250 μM 5-iodo-2′-deoxyuridine (IdU) (Sigma-Aldrich, I7125) for 30 min. Where required, cells were pre-treated with olaparib (10 μM; Selleckchem, S1060), triptolide (1 μM; Sigma-Aldrich, T3652), setanaxib (1 or 10 μM; Merck, S7171), sulforaphane (10 μM; Merck, 574215), or tert-butylhydroquinone (10 μM; Sigma-Aldrich, 112941) as specified in Figure legends. Labeled cells were washed with PBS, harvested by trypsinization, and resuspended in PBS to a concentration of $2.5 \times 10^5$ cells/ml. 2.5 μl of this cell suspension was mixed with 7.5 μl of lysis buffer [200 mM Tris−HCl (pH 7.5), 50 mM EDTA, 0.5% (w/v) SDS] directly on a slide and incubated for 5 min followed by tilting the slides to 30° causing the drops to run down by gravity. The spreads were then air-dried and fixed with methanol/acetic acid (3:1) at RT for 20 min. DNA was denatured in 2.5 M HCl for 1 h at RT and later washed three times with PBS. Slides were blocked in blocking solution [2% (w/v) BSA, 0.1% (v/v) Tween 20 in PBS] for 20 min and incubated with primary antibodies: rat monoclonal anti-BrdU (Abcam, ab6326; 1:500; detection of CldU) and mouse monoclonal anti-BrdU (BD Biosciences, 347580; 1:100; detection of IdU), in blocking solution for 2.5 h at RT in a humid chamber. Afterward, the samples were washed 3 times in PBS-T [0.2% (v/v) Tween 20 in PBS] and incubated with secondary antibodies: Cy3-conjugated donkey anti-rat secondary antibody (ImmunoResearch Europe, 712-166-153; 1:150) and goat anti-mouse Alexa Fluor 488 secondary antibody (Thermo Fisher Scientific, A11001; 1:300), in blocking solution for 1.5 h at RT in a humid chamber. The slides were washed in PBS-T and mounted using Fluoromount-G (Thermo Fisher Scientific, 00-4958-02) mounting media. Images were acquired with a Leica DM6000 upright fluorescent microscope (63 × /1.40 oil immersion). CldU and IdU tract lengths were measured using ImageJ (version 1.54 F).

## S1 nuclease assay

Cells were pulse-labeled with CldU and IdU as described above. Subsequently, cells were washed with PBS and pre-extracted in CSK buffer [25 mM HEPES (pH 7.7), 50 mM NaCl, 1 mM EDTA, 3 mM MgCl₂, 300 mM sucrose, 0.5% (v/v) Triton X-100] for 10 min at RT. Cells were prewashed with S1 buffer [30 mM Sodium Acetate (pH 4.6), 10 mM Zinc Acetate, 5% (v/v) glycerol, 50 mM NaCl] and incubated in S1 buffer with and without S1 nuclease (20 U/ml; Thermo Fisher Scientific, EN0321) for 30 min at 37 °C. After incubation, cells were washed with 0.1% (w/v) BSA in PBS and scraped into an Eppendorf tube. Cells were pelleted by centrifugation at 4600 × g for 10 min. The pellets were resuspended in PBS to a concentration of -1500 cells per μl. 2.5 μl of this cell suspension was mixed with 7.5 μl of lysis buffer [200 mM Tris-HCl (pH 7.5), 50 mM EDTA, 0.5% (w/v) SDS] directly on a slide and incubated for 4 min followed by tilting the slides to 30° causing the drops to run down by gravity. The rest of the procedure was identical to the DNA fiber assay described above.

## Immunofluorescence staining and analysis

For visualization of RNH1(D210N)-GFP foci and γH2AX foci, cells grown on autoclaved coverslips were washed with ice-cold PBS and pre-extracted for 5 min with CSK [25 mM HEPES−NaOH (pH 7.7), 50 mM NaCl, 1 mM EDTA, 3 mM MgCl₂, 0.3 M sucrose, 0.5% (v/v) Triton X-100] buffer on ice. For staining with recombinant RNH1(D210N)-GFP-Tat11 protein, cells were pre-extracted with 0.2% (v/v) Triton X-100 in PBS for 5 min on ice. Subsequently, cells were fixed with 4% (v/v) formaldehyde (Sigma-Aldrich, F8775) in PBS for 15 min in the dark at RT. For PCNA staining, formaldehyde fixation was followed by fixation with ice-cold methanol for 20 min at −20 °C. For 53BP1 staining, coverslips were washed with ice-cold PBS, fixed with 4% (v/v) formaldehyde in PBS for 15 min in the dark at RT, and then permeabilized with 0.1% (v/v) Triton

X-100 for 10 min at RT. After fixations, coverslips were washed with PBS and blocked with 1% (w/v) BSA (Sigma-Aldrich, A7030) in PBS for 10 min at RT. Slides were incubated with primary antibodies at RT or with recombinant RNH1(D210N)-GFP-Tat11 protein at 37 °C for 90 min, subsequently washed with PBS, and incubated with appropriate Alexa Fluor-conjugated secondary antibodies for 30 min. Finally, the coverslips were incubated with DAPI (1 μg/ml; Merck, D9542) and mounted using Fluoromount-G mounting medium (Thermo Fisher Scientific, 00-4958-02). Cells stained with recombinant RNH1(D210N)-GFP-Tat11 protein were mounted with ProLong™ Gold Antifade Mountant (Thermo Fisher Scientific, P36930). The primary antibodies used for immunofluorescence staining: rabbit polyclonal anti-PCNA antibody (Abcam, ab18197; 1:200), rat polyclonal anti-PCNA (Abcam, ab252848; 1:500) mouse monoclonal Cyclin A (Santa Cruz Biotechnology, sc-271682; 1:200), rabbit polyclonal anti-53BP1 (H-300) (Santa Cruz Biotechnology, sc22760; 1:200) and mouse monoclonal anti-γH2AX (Ser139) (Millipore, 05-636; 1:200). The secondary antibodies used for immunofluorescence staining: goat anti-mouse Alexa Fluor 488 IgG (Thermo Fisher Scientific, A11001; 1:400), goat anti-rabbit Alexa Fluor 647 IgG (Thermo Fisher Scientific, A-21245; 1:400), goat anti-rat Alexa Fluor 555 (Thermo Fisher Scientific, A21434; 1:500). Cell images were acquired on a Leica DM6000 upright fluorescent microscope (63 × / 1.40 oil immersion), an IX83 microscope (Olympus) equipped with the ScanR imaging platform using a 60x/1.42 OIL objective with oil immersion, or an IXplore SpinSR10 (Olympus) with a UPLAN S Apo 40x/0.95 NA objective. Images were analyzed with ScanR Analysis software (version 3.5.0) to measure intensity of PCNA, Cyclin A, and with the edge detector plugin to detect RNH1(D210N)-GFP foci, RNH1(D210N)-GFP-Tat11 foci, γH2AX foci, and 53BP1 foci. A minimum of 600 cells per condition were analyzed in most experiments. The DAPI signal was used for segmentation of the images to identify individual nuclei. To prepare scatter plots, the values were exported to the TIBCO Spotfire software (version 10.10.1). An equal number of cells per condition was randomly selected using the Excel RAND() function.

## Proximity ligation assay

For PLA, we used Duolink reagents and followed the Duolink® PLA Fluorescence Protocol (Sigma-Aldrich). Cells grown on coverslips were incubated with 25 μM EdU (Invitrogen, A10044) for 30 min. Cells were pre-extracted and fixed as for immunofluorescence staining. Then cells were blocked with 3% (w/v) BSA in PBS for 10 min. EdU click reaction was performed in buffer containing 100 mM Tris-HCl (pH 8.5), 2 mM CuSO$_4$, 100 mM sodium ascorbate, and 5 μM Alexa Fluor™ 488 Azide (Thermo Fisher Scientific, A10266) for 30 min in the dark followed by two washes with 1% (w/v) BSA in PBS. Coverslips were then incubated with primary antibodies: rabbit polyclonal anti-PCNA antibody (ab18197, Abcam; 1:1000) and mouse monoclonal anti-RNA polymerase II RPB1 (Biolegend, 920204, 1:1000) overnight at 4 °C. Next day, coverslips were washed and incubated with PLA-probes for 1 h at 37 °C: Duolink In Situ PLA Probe Anti Mouse MINUS (Sigma-Aldrich, DUO92004) and Duolink In Situ PLA Probe Anti Rabbit PLUS (Sigma-Aldrich, DUO92002), followed by ligation for 30 min at 37 °C. For amplification, Duolink In Situ Detection Reagents Green (Sigma-Aldrich, DUO92014) was used. Coverslips were then washed and stained with DAPI (1 μg/ml; Merck, D9542) and mounted using Fluoromount-G mounting medium (Thermo Fisher Scientific, 00-4958-02). Cell images were acquired on an IX83 microscope (Olympus) equipped with the ScanR imaging platform using a 60x/1.42 OIL objective with oil immersion. A minimum of 750 cells per condition were analyzed in most experiments. The DAPI signal was used for segmentation of the images to identify individual nuclei. To prepare scatter plots, the values were exported to the TIBCO Spotfire software (version 10.10.1). An equal number of cells per condition was randomly selected using the Excel RAND() function.

## SIRF

U2OS cells growing on autoclaved coverslips were incubated with 25 μM EdU for 10 min, washed twice with ice-cold PBS, pre-extracted with cold 0,2% (v/v) Triton X-100 in PBS for 5 min on ice and fixed with 4% (v/v) formaldehyde solution for 15 min. Subsequently, cells were fixed with methanol for 20 min at −20 °C. EdU click reaction was performed in buffer containing 100 mM Tris-HCl (pH 8.5), 2 mM CuSO$_4$, 100 mM sodium ascorbate, and 100 μM biotin azide (Click Chemistry Tools, 1265-5) for 30 min. PLA was performed using Duolink PLA technology (Sigma-Aldrich) according to the manufacturer's instructions. Cells were incubated with primary antibodies: mouse monoclonal anti-biotin antibody (Jackson ImmunoResearch, 200-002-211, 1:200) and rabbit monoclonal anti-Timeless antibody (Abcam, ab109512, 1:1000) at 4 °C overnight. Next day, coverslips with cells were incubated with PLA probes followed by ligation and amplification according to the manufacturer's protocol. Click reaction to detect EdU was performed in buffer containing 100 mM Tris-HCl (pH 8.5), 2 mM CuSO$_4$, 100 mM sodium ascorbate, and 2 μM Alexa fluor 647 azide (Thermo Fisher Scientific, A10277). Coverslips were stained with DAPI (1 μg/ml; Merck, D9542) and mounted with ProLong™ Gold Antifade Mountant (Thermo Fisher Scientific, P36930). Cells images were acquired on a IXplore SpinSR10 (Olympus) using UPLAN S Apo 40x/ 0.95 NA objective. Images were analyzed with ScanR Analysis software (version 3.5.0). TIMELESS SIRF foci were detected using spot detector. A minimum of 500 cells per condition were analyzed. The DAPI signal was used for segmentation of the images to identify individual nuclei. Mean intensity of EdU was measured to determine S-phase cells.

## 5-EU incorporation assay

Cells were incubated with 1 mM 5-ethynyl uridine (5-EU; Invitrogen, E10345) for 30 min, fixed with 4% (v/v) formaldehyde in PBS for 15 min in the dark at RT and then permeabilized with 0.1% (v/v) Triton X-100 for 10 min at RT. For PCNA staining, formaldehyde fixation was followed by PBS washes and fixation with ice-cold methanol for 20 min at −20 °C. The coverslips were washed with PBS and blocked with 1% (w/v) BSA (Sigma-Aldrich, A7030) in PBS for 10 min at RT. 5-EU click reaction was performed in buffer containing 100 mM Tris-HCl (pH 8.5), 2 mM CuSO$_4$, 100 mM sodium ascorbate, and 5 μM Alexa Fluor™ 488 Azide (Thermo Fisher Scientific, A10266) for 30 min in the dark followed by two washes with 1% (w/v) BSA in PBS. Immunostaining was done with primary rabbit polyclonal anti-PCNA antibody (Abcam, ab18197; 1:200) for 90 min and later with Alexa Fluor 647 goat anti-rabbit (Life Technologies, A21245; 1:400) for 30 min. Finally, the coverslips were incubated with DAPI (1 μg/ml; Merck, D9542) and mounted using Fluoromount-G mounting medium (Thermo Fisher Scientific, 00-4958-02). Cell images were acquired on an IX83 microscope (Olympus) equipped with the ScanR imaging platform using a 60x/1.42 OIL objective with oil immersion. Images were analyzed with ScanR Analysis software (version 3.5.0) to measure intensity of 5-EU and PCNA. Minimum of 700 cells per condition were analyzed. The DAPI signal was used for segmentation of the images to identify individual nuclei. To prepare scatter plots, the values obtained were exported to the TIBCO Spotfire software (version 10.10.1), displaying the same number of cells per condition. An equal number of cells per condition was randomly selected using the Excel RAND() function.

## Slot blot

To detect RNA:DNA hybrids on a slot blot, cells were first harvested by trypsinization, washed with PBS, and centrifuged (1200 × g, 4 °C, 3 min). Pellets were resuspended in lysis buffer [40 mM Tris–HCl (pH 7.5), 1.28 M sucrose, 20 mM MgCl$_2$, 4% (v/v) Triton X-100], incubated for 10 min on ice, and centrifuged (1200 × g, 4 °C, 15 min). The supernatant was removed, and pellets were again resuspended in lysis buffer and centrifuged (1200 × g, 4 °C, 15 min). Isolated nuclei were incubated for 2 h at 50 °C with Proteinase K (0.74 mg/ml;

Roche, 03115852001) in digestion buffer [30 mM Tris-HCl (pH 8.0), 800 mM guanidine, 30 mM EDTA, 5% (v/v) Tween 20, 0.5% (v/v) Triton X-100]. After incubation, the supernatant was mixed with an equal volume of chloroform:isoamyl alcohol (24:1), thoroughly mixed, and centrifuged ($8000 \times g$, 4 °C, 25 min). The aqueous phase was mixed with an equal volume of isopropanol, centrifuged ($8000 \times g$, 4 °C, 10 min) and the precipitated DNA was washed with ethanol and dissolved in TE buffer [10 mM Tris–HCl (pH 8.0), 1 mM EDTA]. Genomic DNA concentration was measured by a NanoDrop. 1.5 μg of DNA was digested or not with 2.4 U of RNase H (NEB, M0297) at 37 °C for 3 h. The DNA samples (750 ng) were spotted in duplicates onto nylon membranes (Hybond-N+; GE Healthcare, RPN203B) using a slot blot manifold (Cleaver Scientific, CSL-S48) and washed with 2xSSC solution [0.3 M sodium chloride, 30 mM trisodium citrate, pH 7.0]. For S9.6 antibody detection, the membrane was dried, UV-crosslinked ($0.12 \text{ J/m}^2$), and after blocking in 5% (w/v) Skimmed Milk Powder (Coop, Switzerland) in TBS-T, incubated with mouse monoclonal S9.6 antibody (Kerafast, ENH001; 1:1000). As a loading control, we used the ssDNA signal. Membrane was dried, denatured in denaturing solution (0.5 M NaOH, 1.5 M NaCl) for 10 min and subsequently neutralized in 0.5 M Tris–HCl (pH 7.2) containing 1.5 M NaCl for 10 min. The membranes were UV-crosslinked ($0.12 \text{ J/m}^2$) and immunostained with mouse monoclonal anti-ssDNA antibody (Millipore, MAB3868; 1:10,000). Both membranes were later incubated with goat anti-mouse IgG-HRP (Sigma-Aldrich, A4416) to detect RNA:DNA hybrids and total DNA, respectively, with chemiluminescence reagents (Pierce ECL Western Blotting Substrate, Thermo Fisher, 32209 and SuperSignal™ West Dura Extended Duration Substrate; Thermo Fisher Scientific, 34075). Signal quantification was performed using ImageJ (version 1.54 F).

### Preparation of cell extracts and western blot analysis

Cells were trypsinized, washed with PBS and resuspended in lysis buffer [50 mM Tris–HCl (pH 7.5), 120 mM NaCl, 0.5% (v/v) Nonidet P-40] supplemented with protease inhibitor cocktail (Roche, 11873580001) and incubated for 5 min on ice. The cell suspension was sonicated with a Diagenode sonicator. Cell lysates were clarified by centrifugation at $16,100 \times g$ for 10 min at 4 °C and protein concentration in cell extracts was determined using the Bradford assay (Bio-Rad). Samples (15–50 μg of total protein) were mixed with Laemmli SDS sample buffer, boiled at 95 °C for 10 min and loaded onto 8–12% SDS-PAGE gel. After electrophoresis, separated proteins were transferred onto a nitrocellulose membrane (VWR international, 10600003) in a wet-transfer apparatus (Bio-Rad, 350 mA, 90 min, 4 °C) with buffer containing 2.5 mM Tris–HCl, 10% (v/v) methanol and 19.2 mM glycine. The membrane was blocked with 3% (w/v) milk in TBS-T [20 mM Tris–HCl (pH 7.5), 150 mM NaCl, 0.1% (v/v) Tween-20] for 30 min at RT and incubated with primary antibodies (diluted in 3% milk/TBS-T) overnight at 4 °C. Afterward, membranes were washed 3 times in TBS-T and incubated with appropriate horseradish peroxidase-coupled (HRP) secondary antibody in TBS-T for 30 min at RT. Membranes were washed in TBS-T and protein bands were detected with a chemiluminescence reagent [Pierce ECL Western Blotting substrate (Thermo Fisher Scientific, 32209), SuperSignal™ West Dura Extended Duration Substrate (Thermo Fisher Scientific, 34075) or SuperSignal™ West Femto Maximum Sensitivity Substrate (Thermo Fisher Scientific, 34094)]. The primary antibodies used for western blotting: HRAS (259) rat monoclonal (Santa Cruz Biotechnology, sc-35; 1:2000); HRAS (C-20) rabbit polyclonal (Santa Cruz Biotechnology, sc-520; 1:500); HRAS rabbit polyclonal (GeneTex, GTX 116041; 1:500), Cyclin E (HE12) mouse monoclonal (Santa Cruz Biotechnology, sc-247; 1:700); TFIIH p89 (S-19) rabbit polyclonal (Santa Cruz Biotechnology, sc-293; 1:1000); RNASE H1 (A-9) mouse monoclonal (Santa Cruz Biotechnology, sc-365783; 1:500); ZRANB3 rabbit polyclonal

(Proteintech, 23111-1-AP; 1:1000); GAPDH (D16H11) rabbit monoclonal (Cell Signaling Technology, 5174; 1:2000); GAPDH (0411) mouse monoclonal (Santa Cruz, sc-47724, 1:1000); SMARCAL1 (D3P5I) rabbit monoclonal (Cell Signaling Technology, 44717S; 1:2000); β-Tubulin (TUB 2.1) mouse monoclonal (Sigma-Aldrich, T4026; 1:1000); MUS81 (B-12) mouse monoclonal (Santa Cruz Biotechnology, sc-376661; 1:500); MUS81 (MTA30 2G10/3) mouse monoclonal (Santa Cruz Biotechnology, sc-53382; 1:500); PRIMPOL rat polyclonal (a gift from Juan Mendez); RECQ1 rabbit polyclonal (Novus Biological, NB100-618; 1:1000); LIG4 (D-8) mouse monoclonal (Santa Cruz Biotechnology, sc-271299; 1:500); ELL (B-4) mouse monoclonal (Santa Cruz Biotechnology, sc-398959; 1:500); PRDX2 (A-2) mouse monoclonal (Santa Cruz Biotechnology, sc-515428; 1:100). The secondary antibodies used for western blotting: goat anti-rabbit IgG-HRP (Sigma-Aldrich, A0545; 1:5000); goat anti-mouse IgG-HRP (Sigma-Aldrich, A4416; 1:5000) and goat anti-rat IgG-HRP (Sigma-Aldrich, A9037; 1:5000).

### Electron microscopy

U2OS T-REx [RNH1-GFP] cells were transduced with HRASV12, cyclin E1 or empty retroviral vectors and either treated with doxycycline (1 ng/ml) for the last 24 h to induce RNaseH1-GFP expression or left untreated. For isolation of genomic DNA, $1.5\text{-}2.7 \times 10^7$ cells were used per sample. All subsequent steps were performed on ice. Cells were pelleted ($600 \times g$, 5 min, 4 °C) and washed with ice-cold PBS. Genomic DNA was then cross-linked by incubating cells with 200 μg/ml trimethylpsoralen for 5 min followed by irradiation with 365-nm UV light for 7 min on a precooled metal plate. This procedure was repeated three times. Cells were then lysed in lysis buffer [1.28 M sucrose, 40 mM Tris-HCl (pH 7.5), 20 mM MgCl₂, 4% (v/v) Triton X-100]. Nuclei were pelleted ($1300 \times g$, 15 min, 4 °C) and incubated for 3 h at 50 °C in 5 ml of digestion buffer [800 mM guanidine-HCl, 30 mM Tris-HCl (pH 8.0), 30 mM EDTA, 5% (v/v) Tween-20, 0.5% (v/v) Triton X-100, 1200 U proteinase K]. Genomic DNA was isolated via chloroform/iso-amylalcohol method, precipitated with isopropanol, and resuspended in TE buffer [10 mM Tris-HCl (pH 8.0), 1 mM EDTA]. Genomic DNA was then digested with PvuII HF (33 U per 10 μg of DNA; NEB, R3151L) plus 33.3 μg of RNase A (Thermo Fisher Scientific, EN0531) and 0.013 U ShortCut RNase III (NEB, M0245S) in CutSmart restriction buffer (3 h, 37 °C) in a total volume of 250 μl. Digested DNA was concentrated and recovered using Microcon DNA Fast Flow Centrifugal Filters (Millipore, MRCF0R100), and DNA concentration was measured. DNA was then spread on a water surface using the benzyldimethylalkylammonium chloride (BAC) method, transferred onto the carbon-coated 400 mesh copper grids and coated with platinum using the Leica EM ACE900 sample preparation system. Grids were examined using the Jeol JEM-1400 Flash transmission electron microscope operated at ≤120 kV (0.2 nm resolution). Images were acquired with a Jeol Flash 2000 × 2000 pixels CMOS camera using the serial EM capturing mode for automated acquisition. Replication forks were analyzed with ImageJ (version 1.54 F). For each sample, ~200 replication forks from three independent biological replicates were analyzed.

### Analysis of bulky anaphase bridges

Cells grown on autoclaved coverslips were synchronized at G2/M phase transition by culturing in the presence of 9 μM RO-3306 (Merck, SML0569) for 16 h. This was followed by three brief washes with PBS for 5 min at RT and subsequent incubation in DMEM medium for a total time of 1.5 h at 37 °C. Cells were fixed with 4% (v/v) formaldehyde in PBS for 15 min at RT, followed by staining with DAPI (1 μg/ml). Cell images were acquired on an IX83 microscope (Olympus) equipped with ScanR imaging platform using a 40 × /0.95 dry objective. The percentage of anaphase cells with bulky bridges was determined using ImageJ (version 1.54 F). At least 25 anaphase cells were scored per condition in each experiment.

## Analysis of micronuclei

Cells were grown on autoclaved coverslips in the presence of the cytokinesis inhibitor cytochalasin B (2 µg/ml, Sigma-Aldrich, C6762) for 16 h, followed by a wash with PBS. Where required, cells were treated with 1 µM setanaxib, 10 µM sulforaphane or 10 µM tert-butylhydroquinone as specified in Figure legends. The cells were washed with PBS and fixed with 4% (v/v) formaldehyde in PBS for 15 min at RT and stained with DAPI (1 µg/ml, Sigma-Aldrich, D9542) for 2 min at RT in the dark. Images were acquired using a Leica DMI 6000 inverted microscope at 20x or 40x magnification or IX81 microscope (Olympus) equipped with a ScanR imaging platform using a 40x/1.3 NA objective with oil immersion. The percentage of binucleated cells with micronuclei was determined using ImageJ (version 1.54 F). At least 100 binucleated cells were scored per condition in each experiment.

## Statistical analysis

Statistical analysis was performed using GraphPad Prism 8 software (version 8.4.3). Details of how data are presented, including the definition of center (mean or median) and error bars, as well as the details of statistical tests for each experiment, including the type of statistical tests used and the number of repeats, can be found in the figure legends. Statistical test results, presented as $p$-values, are shown in the figures. Statistical differences in scatter plots of data from DNA fiber assays, QIBC analysis of 5-EU intensity and total number of PCNA/RNAPII PLA foci, γH2AX foci, 53BP1 foci, TIMELESS SIRF foci, RNH1(D210N)-GFP foci, and RNH1(D210N)-GFP-Tat11 foci were determined by a two-tailed non-parametric Kruskal-Wallis test followed by a Dunn's multiple comparisons test. Statistical differences in the plot of average number of 53BP1 foci were determined by two-tailed paired t-test. Statistical differences for other grouped analyses, i.e., micronuclei, anaphase bridges, average number of γH2AX foci, or frequency of fork reversal were assessed by a repeated-measures one-way ANOVA followed by a Tukey's multiple comparisons test.

## Reporting summary

Further information on research design is available in the Nature Portfolio Reporting Summary linked to this article.

# Data availability

The data supporting the findings of this study are provided in the paper and its Supplementary information. Original microscopy images are too numerous and large to be uploaded to a public repository but are available upon request from the corresponding author. Source data are provided with this paper.

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

## Acknowledgements

We thank Fabrizio d'Adda di Fagagna for pBABEneo-HRASV12 construct, Stefano Ferrari for pLXSNneo-CCNE1 construct, Juan Mendez for PRIM-POL antibody, Eva Petermann for BJ-hTert HRASV12ER-TAM cells and Marcel van Vught for RPE1 CE-TetON cells. We thank to Oldrich Benada and Vlada Filimonenko for help with preparation of samples for EM. We also thank the UZH Center for Microscopy and Image Analysis, the Light Microscopy Core Facility of IMG (MEYS - LM2023050, MEYS - CZ.02.1.01/0.0/0.0/18_046/0016045, RVO – 68378050-KAV-NPUI), and the Electron Microscopy Core Facility of IMG (MEYS - LM2023050, ERDF CZ.02.1.01/0.0/0.0/18_046/0016045, CZ.02.01.01/00/23_015/0008205) for support. This work was supported by grants from the Swiss Cancer League (KFS-5484-02-2022 and KFS-6145-08-2024), the Swiss National Science Foundation (310030_214846), the Czech Science Foundation (25-15542S), Sassella Stiftung, and Stiftung zur Krebsbekämpfung. J.D. was supported by the Czech Science Foundation (21-22593X), H.H. was supported by the Czech Science Foundation (25-15199S). A.Z. was supported by the Charles University Grant Agency (GAUK188724).

## Author contributions

A.O., M.D., A.M., M.A., A.Z., K.S., B.B., V.R., C.K. and J.P. performed the experiments and analyzed the data. M.S. and A.O. performed EM experiments and analyzed the data. J.D., L.M., and H.H. contributed to the design of the experiments. P.J. conceived the study, analyzed the data, and wrote the manuscript with contributions from A.O., M.D. and A.M. All authors revised the manuscript.

## Competing interests

The authors declare no competing interests.
