## [Transparent Peer Review file · Nature Communications]

Distinct mechanisms of replication stress induced by oncogenic RAS and cyclin E1 converge on R-loop-dependent fork reversal

Corresponding Author: Professor Pavel Janscak

Version 0:

Reviewer comments:

Reviewer #1

(Remarks to the Author)

Manuscript: MUS81/PRIMPOL-dependent DNA synthesis counteracts R-loop-induced replication fork reversal to prevent chromosome damage in mitosis upon oncogene activation

The manuscript by Oravetzova et al. presents insights into the mechanisms underlying DNA replication stress induced by overexpression of oncogenic HRASV12 or Cyclin E1. The authors demonstrate that replication stress and DNA damage associated with HRASV12 or Cyclin E1 activation are driven by the accumulation of S-phase-specific R-loops. These R-loops promote replication fork reversal, leading to mitotic DNA damage and micronuclei formation. Supporting this model, the authors show that inhibition of PARP1 or depletion of ZRANB3—both of these conditions, which suppress fork reversal—results in unrestrained fork progression and rescues mitotic DNA damage in cells overexpressing HRASV12 or Cyclin E1. Based on these findings, the authors propose that persistent reversed replication forks are a significant source of mitotic DNA damage during oncogene-induced replication stress. Finally, they show that this unrestrained fork progression in HRASV12- and Cyclin E1-overexpressing cells depends on MUS81 and is accompanied by PRIMPOL-mediated single-stranded DNA gap formation.

Overall, the experiments are generally well-conducted and supported by clear results. However, a key limitation of the manuscript is that the novelty of the insights is somewhat limited. The link between oncogene activation—such as HRASV12 or Cyclin E1—and replication stress, including replication fork slowing and the interplay between global transcription and R-loop formation, is well-established (e.g., DOI: 10.1038/onc.2012.387, DOI: 10.1038/ncomms13087). Similarly, the role of the MUS81–LIG4–PRIMPOL axis in mediating replication restart following R-loop-induced replication stress—though previously studied under different stress conditions—has already been demonstrated by the authors themselves (DOI: 10.1038/ncomms13087) and is here extended to the context of HRASV12- and Cyclin E1-overexpressing cells.

That said, the authors do note an interesting distinction between HRASV12 and Cyclin E1 in their ability to induce global transcription (Supplementary Figure 3b), even though both oncogenes ultimately lead to comparable levels of S/G2-specific R-loops (Figure 2a–c) and a similar rescue of mitotic DNA damage upon R-loop disruption (Figure 1a–h). A more systematic characterization of these differences—particularly in the context of the well-established role of Cyclin E1 in promoting new origin firing—could have provided valuable insights into the distinct upstream mechanisms leading to R-loop formation and replication stress. Such analysis might have clarified how divergent oncogenic triggers can converge on a common outcome—R-loop accumulation—as a key driver of replication stress and genome instability.

Below are some comments and suggestions that authors could improve and strengthen before publication.

Figures 1 and 2:

- In general, the experiments are well-performed. However, the manuscript lacks a comprehensive overview of cell cycle distribution and the extent of DNA damage or replication stress specifically within S-phase.

- Since the authors utilize QIBC in Figure 2, a similar approach could be applied to assess S-phase-specific replication stress markers.
- Along the same lines, it would be informative to evaluate R-loop levels (Figure 2c) upon HRASV12 or Cyclin E1 expression in the presence of CDC7 inhibition. This would help clarify how deregulated origin firing under these conditions influences R-loop formation and could be further extended to include replication fork speed measurements.

Figure 3:

- The authors use ZRANB3 depletion to disrupt fork reversal. It would strengthen the conclusions to test whether similar phenotypes are observed upon depletion of other fork reversal enzymes such as HLF1 or SMARCA1?
- To further elucidate causal relationships, it would be valuable to assess R-loop levels (as shown in Figure 2c) following ZRANB3 or other fork reversal enzyme depletions. This could help determine if R-loops persist or resolve under conditions of oncogene-induced replication stress and unrestrained fork progression.

Figures 4 and 5:

- While the MUS81–LIG4 axis appears to play a critical role in promoting fork progression under oncogene-induced stress conditions, the lack of a comprehensive overview of S-phase DNA damage, global DNA synthesis, and the extent of R-loop accumulation makes it difficult to conceptualize the precise nature and extent of fork stalling that necessitates nucleases like MUS81.
- It is also puzzling that ZRANB3 knockout cells, which replicate with persistent ssDNA gaps, nevertheless show a rescue of mitotic defects typically induced by oncogene-driven replication stress. This raises the question of whether these cells rely on daughter strand gap repair mechanisms, such as RAD51-mediated homologous recombination. This possibility could be addressed by assessing spontaneous RAD51 foci formation during S-phase (e.g., as described in DOI: 10.1016/j.devcel.2021.01.011).

Other points:

- The authors implicate the potential incidence of head-on transcription–replication conflicts as a driver of R-loop formation and replication stress. However, this has not been directly tested. It would be informative to apply genomics-based approaches, for R-loop mapping in relation to replication origin orientation, to explore TRC landscapes in HRASV12 versus Cyclin E1-overexpressing cells. Alternatively, it should be phrased cautiously.
- It is unclear why 5-FU was used to assess global transcriptional activity instead of more conventional and direct methods such as EU (5-ethynyl uridine) incorporation, which is widely used for nascent RNA labelling.

Reviewer #2

(Remarks to the Author)

Oravetzova A. et al. show how an increase of transcriptional levels or replication origin firing mediated by oncogene overexpression are linked to genomic instability associated to R-loop formation in S phase and fork reversal. They also showed that inhibition of fork reversal promotes fork progression re-start and suppression of mitotic DNA damage mediated by MUS81 and PRIMPOL repriming. Reinforcing in the idea that fork reversal have both a protective but also a damaging role if not resolved properly. The manuscript presents a huge amount of work. While the results presented are solid and it is an interesting observation, I do have some concerns regarding the novelty of the presented data.

Major comments

The authors propose that oncogene overexpression (OE) leads to R-loop formation specifically in S phase, possibly due to TRCs, and inducing persistent replication fork reversal. The RNH1 foci seen in G2 phase in Fig2C can represent persistent R-loops formed in S phase as the author suggested but also, R-loop formed in G2 if late replication and TRCs occurs in G2/M phases. Late replication has been shown to occur at TSS regions that is where more R-loops are formed (Wang J. et al Cell Report 2021). This possibility could be tested by for example checking RNH1 foci when inhibiting transcription during S phase and reactivating it during G2, or inhibiting fork reversal resolution and checking whether the frequency increases from S to G2.

Also, to confirm that it is produced as a result of TRC, authors could check directly measuring the frequency of EdU-RNAPII or PCNA-RNAPII interactions by PLA assays after oncogene OE.

The authors also propose that the formation of fork reversal in HRASV12 OE could be due to a slow fork progression mediated by ROS. Their data of fork velocity in the presence of RNH1 OE (Fig1B) support the idea that something else apart from R-loops is delaying the movement of the fork. They could check whether TIMELESS/TIPIN is indeed detached from the replisome and whether this is also the case of Cyclin E OE.

Is ATR active in both oncogene OE conditions?

Reviewer #3

(Remarks to the Author)

In this manuscript, the authors show that HRAS or cyclin E1 overexpression results in enhanced replication fork reversal and increased mitotic DNA damage, which is primarily caused by R-loop mediated transcription-replication conflicts (TR&Cs) during S-phase. Inhibition of replication fork reversal creates a condition of unrestrained replication fork progression which prevents mitotic DNA damage in oncogene overexpressing cells. This process engages the MUS81/LIG4/EEL axis and PRIMPOL-mediated repriming to support unrestrained fork progression, where the presence of persistent reversed forks contributes to enhanced fork slowdown/stalling and an increase in mitotic DNA damage. Overall, experimental evidence is solid and well-controlled. Understanding the nature of DNA replication stress caused by oncogenes is considered significant. Nevertheless, results of this study seem incremental; it is uncertain how much new molecular insight can be deduced from the findings presented here. It is already well-established in the field that (1) oncogene activation, including Cyclin E, causes TRCs, (2) R-loop formation is associated with frequent fork reversal, (3) unrestrained fork progression upon PARP inhibition is dependent the MUS81 activity and the RECCQ1 helicase, and (4) restart of stalled forks, including R-loops, via MUS81 requires repriming mediated by PRIMPOL, all of which are the key takeaways of this study. In this case, overexpression of oncogenes is used to trigger DNA replication stress - unless new mechanistic insight of how HRASG12V and Cyclin E uniquely compromises replication fork stability and causes R-loop is presented, the current study lacks sufficient molecular details and is only limited to the phenotypic analyses using loss-of-function approaches (e.g., siRNA or knockout).

1. One major concern is that the phenotypic analyses are limited to measuring DNA fiber tracks and there is little molecular insight to explain mitotic DNA damage. This is a major weakness of the manuscript in its current form.
2. Figure 1 to establishes R-loop as a source of oncogene-induced replication stress is performed only in a cancer cell line (U2OS), which has already undergone a transformation process. On the other hand, Fig. S4 employs immortalized yet non-transformed fibroblast or epithelial cells. The authors should at least confirm the key results of Figure 1 using non-transformed cells.
3. In Figure 1, it should be tested whether RNaseH1 expression can suppress DNA damage markers (e.g., H2AX). In relation to this, the nature of DNA lesions (e.g., the status of DSBs) has not been scrutinized. What would be the underlying basis of DNA damage that contributes to defective chromosome segregation and mitotic DNA damage after R-loop accumulation?
4. The authors uses RNaseH1-dependent rescues of fork progression as sole evidence to support that R-loops are the cause of oncogene-induced replication stress. However, there is no direct evidence of R-loop formation in these oncogene overexpressing cells. This could be validated by using the S9.6 antibody previously used by the lab in Ref#25 either through immunofluorescence or DRIP-seq. A positive control such as PDS or even HU or APH can be included as mentioned by the authors.
5. 5-FU staining suggests that there is an increase in transcriptional activity in G1, S, and G2/M cells that are overexpressing HRASG12V but not in cyclin E1 overexpressing cells, while the authors claim that R-loops seen in S-phase are carried over into mitosis. Can the authors exclude the possibility of R-loop formation within G2/M itself independently of DNA replication?
6. The authors show that ZRANB3 completely prevents oncogene-induced anaphase bridges and micronuclei (Fig. 3). It is an interesting result since other translocases such as SMARCAL1 or HLTF are also known to be involved in fork reversal. These factors should be tested to determine how much individual translocase contributes to the fork reversal process. If solely dependent on ZRANB3, some mechanistic explanation should be presented considering the way ZRANB3 acts to promote fork reversal.
7. It would be important to test whether PRIMPOL's primase activity is actually required for unrestrained replication fork progression.
8. Figure 6 model – there is no experimental evidence supporting that ZRANB3 deficiency rescues the mitotic defect. In relation to this, the physiological relevance of inhibiting fork reversal is not clear. The phenotype of unrestrained DNA synthesis is revealed when cells are forced to lose fork reversal process, and whether unrestrained DNA synthesis plays a meaningful role in normal DNA replication and mitosis is not addressed. Additionally, there is no mechanism of fork remodeling and processing that is uniquely engaged by oncogenes.

Version 1:

Reviewer comments:

Reviewer #1

(Remarks to the Author)

The revised manuscript by Oravetzova, Dvorakova et al. includes substantial additional work and improved clarity compared to the initial submission. As noted previously, the study is technically well executed, supported by complementary

experimental approaches and appropriate statistical analyses. In the original version, however, the novel mechanistic insights were somewhat limited.

In the revised manuscript, the authors now more effectively develop the concept of mechanistic convergence in oncogene-induced replication stress, focusing on HRASV12 and Cyclin E1. In particular, the clarification that the ROS–PRDX2 axis acts as an upstream driver of replication fork slowing, and that it contributes to the formation of transcription–replication conflicts (TRCs) in HRASV12 but not in Cyclin E1, strengthens the overall contribution of the study. The additional data, including the use of NOX inhibitors, ROS quenching, and PRDX2 depletion experiments, provide useful insight into how distinct oncogenic programs can lead to overlapping replication stress outcomes.

The authors have adequately addressed the specific comments raised during revision, and the revised manuscript presents a large amount of data more systematically and coherently, improving its overall impact for understanding how oncogenes shape replication stress and genome stability. One minor comment from my side is that the authors should consider using a more specific title that better reflects the key data and conclusions presented in the study, rather than relying on a broad and generic title.

Reviewer #2

(Remarks to the Author)

I appreciate the authors' substantial effort to improve the quality and novelty of the manuscript. The revised version is clearly stronger than the original one, and the inclusion of additional mechanistic data regarding the role of ROS and PRDX2 in the replication stress induced by HRASV12 has significantly improved the novelty of the study. Nevertheless, I still have some concerns that should be addressed.

The authors conducted new PLA experiments to obtain evidence that overexpression of HRASV12 or cyclin E1 increases the frequency of head-on TRCs. Although they acknowledge that PLA data do not provide direct evidence for the link between TRCs and R-loop, and they have turned down the statements they claimed that the increase of PLA foci when overexpressing HRASV12 is linked to HO TRCs. They based their conclusions on the observation made by Hamperl et al., 2017 where head-on but not co-directional TRCs increase PLA foci between RNAPII and PCNA. However, these results have been obtained from artificial episomal systems. A recent study has demonstrated that co-directional TRCs, where the RNAPII is which collides with the replication fork traveling ahead, also exists (Bruno et al Mol Cell 2024; doi:10.1016/j.molcel.2023.11.036). These new type of TRCs that should not block replication fork progression can also be detected by these PLA experiments. Consequently, this possibility should be discussed, and it should be acknowledged that at least a fraction of the increase in PLA foci may not correspond to head-on (HO) TRCs. This is especially relevant in the context of HRASV12 overexpression that leads to an increase in transcriptional activity.

Moreover, to clearly determine whether HO TRCs are responsible for the observed genomic and mitotic instability, the manuscript would greatly benefit from the inclusion of the HRASV12 overexpression CUT&Tag genomic data the authors are generating and the estimation of replication fork directionality that can be done using public U2OS Repli-seq data. Alternatively, they should be more cautious when assigning directionality to TRCs.

Reviewer #3

(Remarks to the Author)

The authors addressed many concerns raised in the previous version and strengthened the manuscript. The revised title does not summarize their original findings, though. It is recommended to come up with a more specific title.

Point-by-point response to the reviewers' comments (NCOMMS-25-21283A)

We would like to thank the reviewers for taking the time to carefully assess our manuscript and for providing helpful and constructive feedback that helped us significantly improve the manuscript.

Answers to specific questions of reviewers:

Reviewer #1 (Remarks to the Author):

Manuscript: MUS81/PRIMPOL-dependent DNA synthesis counteracts R-loop-induced replication fork reversal to prevent chromosome damage in mitosis upon oncogene activation

The manuscript by Oravetzova et al. presents insights into the mechanisms underlying DNA replication stress induced by overexpression of oncogenic HRASV12 or Cyclin E1. The authors demonstrate that replication stress and DNA damage associated with HRASV12 or Cyclin E1 activation are driven by the accumulation of S-phase-specific R-loops. These R-loops promote replication fork reversal, leading to mitotic DNA damage and micronuclei formation. Supporting this model, the authors show that inhibition of PARP1 or depletion of ZRANB3—both of these conditions, which suppress fork reversal—results in unrestrained fork progression and rescues mitotic DNA damage in cells overexpressing HRASV12 or Cyclin E1. Based on these findings, the authors propose that persistent reversed replication forks are a significant source of mitotic DNA damage during oncogene-induced replication stress. Finally, they show that this unrestrained fork progression in HRASV12- and Cyclin E1-overexpressing cells depends on MUS81 and is accompanied by PRIMPOL-mediated single-stranded DNA gap formation.

Overall, the experiments are generally well-conducted and supported by clear results. However, a key limitation of the manuscript is that the novelty of the insights is somewhat limited. The link between oncogene activation—such as HRASV12 or Cyclin E1—and replication stress, including replication fork slowing and the interplay between global transcription and R-loop formation, is well-established (e.g., DOI: 10.1038/onc.2012.387, DOI: 10.1038/ncomms13087). Similarly, the role of the MUS81–LIG4–PRIMPOL axis in mediating replication restart following R-loop-induced replication stress—though previously studied under different stress conditions—has already been demonstrated by the authors themselves (DOI: 10.1038/ncomms13087) and is here extended to the context of HRASV12- and Cyclin E1-overexpressing cells.

That said, the authors do note an interesting distinction between HRASV12 and Cyclin E1 in their ability to induce global transcription (Supplementary Figure 3b), even though both oncogenes ultimately lead to comparable levels of S/G2-specific R-loops (Figure 2a–c) and a similar rescue of mitotic DNA damage upon R-loop disruption (Figure 1a–h). A more systematic characterization of these differences—particularly in the context of the well-established role of Cyclin E1 in promoting new origin firing—could have provided valuable insights into the distinct upstream mechanisms leading to R-loop formation and replication stress. Such analysis might have clarified how divergent oncogenic triggers can converge on a

common outcome—R-loop accumulation—as a key driver of replication stress and genome instability.

Response: We would like to thank the reviewer for valuable suggestions on how to improve our manuscript. To address the reviewer's issues regarding limited novelty, we have included, in the revised manuscript, the results of our follow-up project aiming to clarify the molecular mechanism leading to R-mediated fork stalling in HRASV12-expressing cells. As mentioned in our manuscript, apart from increasing global transcription, oncogenic RAS is known to increase the cellular levels of reactive oxygen species (ROS) by upregulating NADPH oxidases 1 and 4 (Kodama et al., 2013; Mitsushita et al., 2004; Ogrunc et al., 2014; Weyemi et al., 2012). Work in Jiri Lukas lab has shown that excessive ROS disrupt the oligomeric state of the replisome-associated ROS sensor peroxiredoxin 2 (PRDX2) and thereby enforce dissociation of the TIMELESS-TIPIN complex from the replisome, leading to a global reduction in replication fork velocity (Somyajit et al., 2017). Moreover, our recent study revealed that this ROS-induced fork slowdown ultimately causes R-loop-mediated fork stalling at sites of head-on TRCs (Andrs et al., 2023). We therefore asked whether ROS play a role in R-loop-mediated replication stress induced by oncogenic RAS. Our experiments revealed that HRASV12-induced replication stress is largely driven by ROS in a manner dependent on PRDX2 and is linked to PRDX2-mediated release of the fork acceleration factor TIMELESS from the replication fork. On the other hand, replication fork stalling in cyclin E1-overexpressing cells was found to be ROS- and PRDX2-independent, and is presumably a consequence of increased intragenic origin firing as reported previously (Macheret and Halazonetis, 2018). These new findings, presented in Fig. 6-8, and Supplementary Fig. 11-13, and described on pages 11-13, provide explanation for our observation that HRASV12 induces R-loop formation in S phase but not in G1, although it increases transcription at all stages of the cells cycle. Moreover, these findings argue against the model wherein HRASV12-induced replication stress stems from increased global transcription due to upregulation of TATA-box binding protein (TBP), which increases R-loop formation (Kotsantis et al., 2016). Instead, we propose that oncogenic RAS induces R-loop-mediated fork staling by increasing the cellular levels of ROS, which in turn induce TIMELESS-TIPIN dissociation from the replisome, leading a reduction in replication fork velocity. Conflicts of such slowly moving forks with head-on transcription complexes lead to R-loop formation which induces fork reversal and thereby halts fork progression. In support of this model, we also show that HRASV12-induced replication stress was rescued by NOX1/4 inhibition (Fig. 6a, and Supplementary Fig. 11b). The rescue of HRASV12-induced replication stress by TBP depletion reported by Eva Petermann lab could be explained by downregulation of NOX1/4 enzymes, which generate ROS in HRASV12-expressing cells. This would be expected in case of a deficiency in a general transcription factor such as TBP. Nevertheless, we cannot exclude the possibility that increased transcription activity enhances R-loop formation at sites of head-on TRCs in HRASV12-expressing cells. This is now mentioned in Discussion (page 13/14). In any case, we think that our new findings provide important insight into the molecular mechanism underlying DNA replication stress induced by oncogenic RAS. Please note that, due to the addition of these new results, we have changed the title of the manuscript to "*Mechanistic insights into oncogene-induced replication stress*".

Below are some comments and suggestions that authors could improve and strengthen before publication.

Figures 1 and 2:

- In general, the experiments are well-performed. However, the manuscript lacks a comprehensive overview of cell cycle distribution and the extent of DNA damage or replication stress specifically within S-phase.

- Since the authors utilize QIBC in Figure 2, a similar approach could be applied to assess S-phase-specific replication stress markers.

Response: We tested the effect of HRASV12 and cyclin E1 overexpression on the formation of γ H2AX and 53BP1 foci in different stages of the cell cycle. By QIBC analysis, we found that overexpression of either oncogene increased γ H2AX foci in S-phase cells identified by PCNA staining. Notably, only a mild increase in 53BP1 foci - marker of DNA double-strand breaks - was detected in S/G2 cells (cells positive for cyclin A staining) upon overexpression of HRASV12 or cyclin E1, suggesting that replication fork stalling induced by these oncogenes does not lead to extensive DNA breakage during S-phase. Additionally, a modest increase in 53BP1 foci was detected in G1 cells upon overexpression of either oncogene, potentially reflecting unresolved under-replicated DNA from the previous cell cycle that led to DNA breakage during chromosome segregation (Ying et al., 2013). This is consistent with the finding that overexpression of these oncogenes increased the frequency of anaphase bridges and micronuclei (Fig. 1g, h; Supplementary Fig. 1e, f). Please note that the data from our analysis of γ H2AX and 53BP1 foci are now presented in Supplementary Fig. 2 and described on page 6.

As suggested by the reviewer, we also tested the effect of HRASV12 or cyclin E1 overexpression on cell cycle distribution. We did not observe any major differences in cell cycle distribution compared to control cells except for a small reduction in the proportion of G1 cell in cyclin E1-overexpressing cell populations, and a small reduction in the proportion of S-phase cells in HRASV12-overexpressing cell population. The former finding is consistent with the finding of Halazonetis lab that cyclin E1 overexpression shortens the length of G1 phase. We think that it is not necessary to include these data in the manuscript and therefore we show them only in this response letter (please see Fig. R1 below).

Figure R1. Cell cycle profile of U2OS cells transduced with HRASV12 (RAS) or cyclin E1 (CE) retroviral vector. Asynchronously growing cells were treated with EdU for 30 min prior to harvest for FACS analysis to determine actively replicating cells. DAPI counterstaining was used to follow DNA content. EV, empty vector. Data are mean +/- SD, n = 4

- Along the same lines, it would be informative to evaluate R-loop levels (Figure 2c) upon HRASV12 or Cyclin E1 expression in the presence of CDC7 inhibition. This would help clarify how deregulated origin firing under these conditions influences R-loop formation and could be further extended to include replication fork speed measurements.

Response: We tested the effect of CDC7 inhibition on R-loop accumulation in HRASV12-overexpressing cells by treating cells with 2 μ M XL413 for 2 h followed by fixation and staining cells with recombinant RNH1(D210N)-GFP protein. QIBC analysis showed that HRASV12 overexpression increased R-loops (RNH1 foci), but this was not significantly affected by CDC7 inhibition (please see Fig. R2 below). We also tested the effect of CDC7 inhibition on replication fork progression in cells overexpressing HRASV12 or cyclin E1. In this experiment, XL413 was present during DNA fiber labeling with CldU and IdU. We found that CDC7 inhibition could rescue both HRASV12- and cyclin E1-induced replication fork slowing. These results are now shown in Supplementary Fig. 14 and discussed in the manuscript on page 14.

Figure R2. *Top:* Experimental workflow. Doxycycline (DOX; 1 μ g/ml) was added for the last 48 h to induce HRASV12 expression. Cells were treated with 2 μ M CDC7i (XL413) for 2 h. *Bottom:* Quantification of the number of RNH1(D210N)-GFP-Tat11 foci per PCNA-positive nucleus for the indicated conditions ($n \geq 401$). Black horizontal lines indicate the mean; p -values were calculated by the Kruskal-Wallis test followed by Dunn's multiple comparisons test.

Figure 3:

- The authors use ZRANB3 depletion to disrupt fork reversal. It would strengthen the conclusions to test whether similar phenotypes are observed upon depletion of other fork reversal enzymes such as HLF1 or SMARCAL1?

Response: We tested the effect of SMARCAL1 depletion on replication fork slowing and micronucleation induced by HRASV12 or cyclin E1 overexpression. We observed similar phenotypes as in case of ZRANB3 depletion. These data are now shown in Supplementary Fig. 7c, d, j.

- To further elucidate causal relationships, it would be valuable to assess R-loop levels (as shown in Figure 2c) following ZRANB3 or other fork reversal enzyme depletions. This could help determine if R-loops persist or resolve under conditions of oncogene-induced replication stress and unrestrained fork progression.

Response: We tested the effect of ZRANB3 depletion on R-loop accumulation in HRASV12-overexpressing U2OS cells. R-loops were detected by staining cells with recombinant RNH1(D210N)-GFP protein followed by QIBC analysis. We found that upon ZRANB3 depletion, HRASV12 overexpression did not induce accumulation of RNH1(D210N)-GFP foci in S-phase cells. This is consistent with the model wherein R-loop resolution is a prerequisite for replication restart via MUS81-LIG4 axis (Rao et al., 2024). The data showing that ZRANB3

depletion suppresses R-loop accumulation in HRASV12-overexpressing cells are now shown in Supplementary Fig. 12d-f.

Figures 4 and 5:

- While the MUS81–LIG4 axis appears to play a critical role in promoting fork progression under oncogene-induced stress conditions, the lack of a comprehensive overview of S-phase DNA damage, global DNA synthesis, and the extent of R-loop accumulation makes it difficult to conceptualize the precise nature and extent of fork stalling that necessitates nucleases like MUS81.

Response: Our aim was to determine whether the MUS81-LIG4-PRIMPOL axis is required for unrestrained DNA synthesis induced by inhibition of fork reversal in oncogene expressing cells. We believe that the DNA fiber data shown in Fig. 4 and 5 strongly support this claim.

Measurement of global DNA synthesis for example by quantifying EdU incorporation cannot provide accurate information about replication fork stalling events because it also scores for dormant origin firing. We believe that single molecule approaches such as DNA fiber assay are more appropriate to study the processes occurring at stalled replication forks. To support our claim that MUS81 is essential for restarting stalled replication forks in HRASV12/cyclin E1-overexpressing cells, we reanalyzed DNA fibers in Fig. 4b to determine sister fork ratio, a measure of fork stalling frequency, for each condition. The obtained data, which are shown below in Fig. R3, demonstrate that the rescue of sister fork asymmetry (fork stalling) in HRASV12/cyclinE1-overexpressing cells by ZRANB3 depletion depends on MUS81 but not on RECQ1.

Figure R3. MUS81 is required for the rescue of oncogene-induced replication fork stalling by ZRANB3 depletion. *Left:* Workflow for DNA fiber labeling in transduced U2OS cells 3 days after selection. Cells were transfected with appropriate siRNAs on day 0. *Right:* Plot of the values of IdU tract length ratio of sister forks (sister fork ratio) ($n \geq 100$). Black horizontal lines indicate the median; p -values were calculated by the Kruskal-Wallis test followed by Dunn’s multiple comparisons test.

- It is also puzzling that ZRANB3 knockout cells, which replicate with persistent ssDNA gaps, nevertheless show a rescue of mitotic defects typically induced by oncogene-driven replication stress. This raises the question of whether these cells rely on daughter strand gap repair mechanisms, such as RAD51-mediated homologous recombination. This possibility could be addressed by assessing spontaneous RAD51 foci formation during S-phase (e.g., as described in DOI: 10.1016/j.devcel.2021.01.011).

Response: Increased frequency of mitotic errors (anaphase bridges and micronuclei) in oncogene-expressing cells results from incompletely replicated DNA regions caused by persistent reversed forks. It is likely that ssDNA gaps generated during PRIMPOL-mediated restart of R-loops-stalled forks in oncogene-expressing cells lacking ZRANB3 are filled by homologous recombination before the onset of mitosis. In any case, it is not expected that persistence of such ssDNA gaps would impair chromosome segregation in mitosis giving rise to anaphase bridges and micronuclei. As suggested, we tried to monitor these putative HR events by measuring RAD51 foci during S-phase. However, we found that overexpression of HRASV12 or cyclin E1 did not significantly increase the frequency of RAD51 foci in S-phase cells even upon depletion of ZRANB3 (please see Fig. R4 below). Obviously, this does not exclude the possibility that the ssDNA gaps are eliminated by HR. It is also possible that the ssDNA gaps are simply filled in by DNA polymerases as the DNA template does not contain obstacles that would block DNA synthesis as in case PRIMPOL-mediate repriming at bulky DNA adducts (e.g. cisplatin crosslinks).

Figure R4. a Top: Experimental workflow. U2OS HRASV12 TetON cells were transfected with siLUC or siZRANB3 three days before fixation. Doxycycline (DOX; 1 μ g/ml) was added for the last 48 h to induce HRASV12 expression. **Bottom:** Quantification of the number of RAD51 foci per PCNA-positive nucleus for the indicated conditions (n = 760). **b Top:** Experimental workflow. U2OS CE TetON cells were transfected with siLUC or siZRANB3 three days before fixation. Doxycycline (DOX; 1 μ g/ml) was added for the last 48 h to induce cyclin E1 (CE) expression. **Bottom:**

Quantification of the number of RAD51 foci per PCNA positive nucleus for the indicated conditions (n = 640). **a, b** A representative plot from two independent experiments yielding similar results is shown. Black horizontal lines indicate the median; *p*-values were calculated by the Kruskal-Wallis test followed by Dunn's multiple comparisons test.

Other points:

- The authors implicate the potential incidence of head-on transcription–replication conflicts as a driver of R-loop formation and replication stress. However, this has not been directly tested. It would be informative to apply genomics-based approaches, for R-loop mapping in relation to replication origin orientation, to explore TRC landscapes in HRASV12 versus Cyclin E1-overexpressing cells. Alternatively, it should be phrased cautiously.

Response: To obtain evidence that overexpression of HRASV12 or cyclin E1 increases the frequency of head-on TRCs, we measured colocalization between RNAPII and PCNA by proximity ligation assay (PLA). Please note that work in Karlene Cimprich lab has shown that head-on but not co-directional TRCs increase PLA foci between RNAPII and PCNA (Hamperl et al., 2017). We found that overexpression of both HRASV12 and cyclin E1 increases the frequency of RNAPII/PCNA PLA foci in U2OS cells, indicating head-on TRCs. These data are

now shown in Supplementary Fig. 4. Nevertheless, as suggested by the reviewer, we have tuned down our statements in the manuscript because the above PLA data do not provide direct evidence for the link between TRCs and R-loop formation.

We would like to mention here that we have recently initiated experiments to map the sites of R-loop formation in HRASV12-overexpressing cells by Cut&Tag technology. In the first experiment, we identified 34,607 R-loop peaks for the empty vector (EV) sample and 51,634 R-loop peaks for the HRASV12 sample. As expected, the most significant peaks of R-loops were observed at promoters/transcription start sites for both conditions (47% for EV, 43% for HRASV12). However, we also found a large number of intragenic peaks that were enriched in the HRASV12 sample (12043 for EV vs. 19827 for HRASV12). These peaks were predominantly unique and did not overlap between the two conditions. We also analyzed the locations of R-loops relative to the origins of DNA replication. To do this, we used publicly available data from SNS-seq performed by Guilbaut et al. 2022 (DOI: 10.1093/nar/gkac555). We selected replication origins within the genes on the (+) strand. We detected the presence of R-loops not only at the sites of the origins but also approximately 3 to 4 kb upstream of the origins. This phenomenon was particularly evident in the regions of common fragile sites (Figure R5a), which are known to be highly prone to breakage upon HRASV12-induced replication stress. Importantly, no such peaks were observed downstream of the intragenic origins or at the intergenic origins, supporting the notion that HRASV12-induced R-loops form at sites of head-on TRCs (Figure R4b). Obviously, these are very preliminary data that need to be confirmed by at least one more Cut&Tag experiment. Therefore, we cannot include these data in the current manuscript. These data will be part of a follow-up manuscript, where we also intend to perform Cut&Tag experiments to map the location of stalled forks (MCM7, pol epsilon) and transcription complexes (elongating form of RNAPII) not only for HRASV12 but also for cyclin E1 overexpression to obtain a complete genome-wide profile of the sites of R-loop-mediated fork stalling induced by these oncogenes.

Figure R5. R-loops form at sites of head-on transcription-replication conflicts in HRASV12-overexpressing cells. (A) S9.6-CUT&Tag-Seq signal across intragenic origins for forward oriented genes (+ strand) within common fragile sites (CFSs). (B) S9.6-CUT&Tag-Seq signal across intergenic origins. (A and B) Yellow line: U2OS cells overexpressing HRASV12; Blue line, U2OS cells harboring empty vector; Dark purple line: U2OS cells harboring empty vector and treated with RNaseA prior to CUT&Tag assay. The data show that R-loops accumulate on the head-on side of origins in gene bodies

- It is unclear why 5-FU was used to assess global transcriptional activity instead of more conventional and direct methods such as EU (5-ethynyl uridine) incorporation, which is widely used for nascent RNA labelling.

Response: As suggested, we used 5-ethynyl uridine to label nascent RNA in oncogene overexpressing cells and obtained essentially the same results as with 5-fluorouridine. The new data are shown in Supplementary Fig. 6.

Reviewer #2 (Remarks to the Author)

Oravetzova A. et al. show how an increase of transcriptional levels or replication origin firing mediated by oncogene overexpression are linked to genomic instability associated to R-loop formation in S phase and fork reversal. They also showed that inhibition of fork reversal promotes fork progression re-start and suppression of mitotic DNA damage mediated by MUS81 and PRIMPOL repriming. Reinforcing in the idea that fork reversal have both a protective but also a damaging role if not resolved properly. The manuscript presents a huge amount of work. While the results presented are solid and it is an interesting observation, I do have some concerns regarding the novelty of the presented data.

Response: We are pleased that the reviewer found our study interesting. We believe that the new findings presented in the revised manuscript, particularly those providing the mechanistic insights into the role of ROS in HRASV12-induced replication stress, will allay the reviewer's concerns about the novelty of the data (for further details, please refer to our response to Reviewer 1 on page 2).

Major comments

The authors propose that oncogene overexpression (OE) leads to R-loop formation specifically in S phase, possibly due to TRCs, and inducing persistent replication fork reversal. The RNH1 foci seen in G2 phase in Fig2C can represent persistent R-loops formed in S phase as the author suggested but also, R-loop formed in G2 if late replication and TRCs occurs in G2/M phases. Late replication has been shown to occur at TSS regions that is where more R-loops are formed (Wang J. et al Cell Report 2021). This possibility could be tested by for example checking RNH1 foci when inhibiting transcription during S phase and reactivating it during G2, or inhibiting fork reversal resolution and checking whether the frequency increases From S to G2.

Response: Wang et al. (2021) show that a subset of transcription start sites (TSSs) are replicated in G2/M due to high levels of antisense transcription, making them difficult to duplicate during S phase. It is possible that some R-loops may result from TRCs at sites of late replication in G2. However, it is likely that these head-on TRCs occur already in S-phase and prevent replication due to R-loop formation. To monitor head on TRCs in HRASV12/cyclin E1-overexpressing cells, we performed PLA between elongating form of RNAPII and PCNA (Hamperl et al., 2017). By QIBC analysis, we found that overexpression of either oncogene increased RNAPII/PCNA PLA foci in S-phase cells but not in G2 cells, suggesting that RNH1 foci seen in G2 cells rather reflect R-loops generated in S-phase. These data are now shown in Supplementary Fig. 4 and described on page 7.

We are not sure whether the experiments suggested by the reviewer would yield conclusive results. It is challenging to establish experimental conditions to eliminate transcription or fork reversal selectively during the entire S phase. It is also likely that prolonged inhibition of transcription would have a negative impact on cell cycle progression.

Also, to confirm that it is produced as a result of TRC, authors could check directly measuring the frequency of EdU-RNAPII or PCNA-RNAPII interactions by PLA assays after oncogene OE.

Response: As mentioned above, we found that overexpressed oncogenes increase RNAPII/PCNA PLA foci in S-phase, but not G2 cells (Supplementary Fig. 4). This suggests that TRCs only occur in S, and not in G2.

The authors also propose that the formation of fork reversal in HRASV12 OE could be due to a slow fork progression mediated by ROS. Their data of fork velocity in the presence of RNH1 OE (Fig1B) support the idea that something else apart from R-loops is delaying the movement of the fork. They could check whether TIMELESS/TIPIN is indeed detached from the replisome and whether this is also the case of Cyclin E OE.

Response: To determine whether HRASV12 overexpression induces dissociation of the TIMELESS/TIPIN complex from the replisome, we used a PLA-based assay monitoring colocalization between TIMELESS and replication sites labeled by EdU incorporation into nascent DNA. QIBC analysis revealed that HRASV12 overexpression significantly reduced the number of TIMELESS/EdU PLA foci in nuclei of U2OS cells in a manner dependent on the replisome-associated ROS sensor PRDX2. On the contrary, upon cyclin E1 overexpression, only a marginal reduction in the number of TIMELESS/EdU PLA foci was observed in PRDX2-proficient cells, with a further decrease upon PRDX2 depletion. These data, which are now shown in Fig. 8 and Supplementary Fig. 13e, f, support the conclusion that R-loop-mediated fork stalling induced by oncogenic RAS is a consequence of ROS-induced dissociation of the TIMELESS/TIPIN complex from the replisome, which results in reduced fork velocity.

Is ATR active in both oncogene OE conditions?

Response: We observed that overexpression of both oncogenes induced γ H2AX formation in S-phase cells, without causing extensive DNA breakage, which suggests ATR activation by stalled forks (please see Supplementary Fig. 2). Nevertheless, we did not observe a pronounced increase in the phosphorylation of ATR targets such Chk1 Ser317 and Chk1 Ser345 upon overexpression of HRASV12 or cyclin E1 in U2OS cells (data not shown).

Reviewer #3 (Remarks to the Author)

In this manuscript, the authors show that HRAS or cyclin E1 overexpression results in enhanced replication fork reversal and increased mitotic DNA damage, which is primarily caused by R-loop mediated transcription-replication conflicts (TR&Cs) during S-phase. Inhibition of replication fork reversal creates a condition of unrestrained replication fork progression which prevents mitotic DNA damage in oncogene overexpressing cells. This process engages the MUS81/LIG4/EEL axis and PRIMPOL-mediated repriming to support unrestrained fork progression, where the presence of persistent reversed forks contributes to enhanced fork slowdown/stalling and an increase in mitotic DNA damage. Overall, experimental evidence is solid and well-controlled. Understanding the nature of DNA replication stress caused by oncogenes is considered significant. Nevertheless, results of this study seem incremental; it is uncertain how much new molecular insight can be deduced from the findings presented here. It is already well-established in the field that (1) oncogene activation, including Cyclin E, causes TRCs, (2) R-loop formation is associated with frequent

fork reversal, (3) unrestrained fork progression upon PARP inhibition is dependent the MUS81 activity and the RECCQ1 helicase, and (4) restart of stalled forks, including R-loops, via MUS81 requires repriming mediated by PRIMPOL, all of which are the key takeaways of this study. In this case, overexpression of oncogenes is used to trigger DNA replication stress - unless new mechanistic insight of how HRASG12V and Cyclin E uniquely compromises replication fork stability and causes R-loop is presented, the current study lacks sufficient molecular details and is only limited to the phenotypic analyses using loss-of-function approaches (e.g., siRNA or knockout).

Response: We are pleased that the reviewer found our results to be solid and well-controlled. We agree that the impact of our study would be further enhanced by providing insights into how HRASV12 and cyclin E1 induce R-loop-mediated fork slowing. In response, we have incorporated data from our follow-up project that specifically addressed this question. We kindly ask the reviewer to refer to our response to Reviewer 1, where we summarized our findings (Page 2).

1. One major concern is that the phenotypic analyses are limited to measuring DNA fiber tracks and there is little molecular insight to explain mitotic DNA damage. This is a major weakness of the manuscript in its current form.

Response: We have realized that the term “*mitotic DNA damage*” has been inappropriately used in our manuscript. We show that both oncogenes increase the frequency of anaphase bridges and micronuclei in mitosis. It has been demonstrated that these phenotypes, within the context of DNA replication stress, are the result of underreplicated DNA regions, which impair chromosome segregation (Ying et al., 2013). Our data show that anaphase bridges and micronuclei induced by overexpression of HRASV12 or cyclin E1 are caused by R-loops (rescue by RNH1 overexpression) and persistent reversed forks (rescue by ZRANB3/SMARCAL1 depletion). Thus, we believe that our findings provide molecular insight into how the studied oncogenes impair chromosome segregation and thereby induce chromosomal instability. We have modified the manuscript text accordingly.

2. Figure 1 to establishes R-loop as a source of oncogene-induced replication stress is performed only in a cancer cell line (U2OS), which has already undergone a transformation process. On the other hand, Fig. S4 employs immortalized yet non-transformed fibroblast or epithelial cells. The authors should at least confirm the key results of Figure 1 using non-transformed cells.

Response: We agree with the reviewer that non-transformed cells are more appropriate for investigating the mechanisms underlying oncogene-induced replication stress. To address this point, we utilized primary BJ fibroblasts with inducible expression of HRASV12 or cyclin E1 (Maya-Mendoza et al., 2015). Similar to U2OS cells, we observed that replication fork stalling induced by HRASV12 or cyclin E1 in these cells depends on RNAPII transcription (rescue by RNAPII inhibition with triptolide; Supplementary Fig. 11b, c) and is alleviated by PARP1 inhibition in a MUS81-dependent manner (Supplementary Fig. 7g, h, and Supplementary Fig. 9b, d). Furthermore, in these cells, we could reproduce our new findings that HRASV12, but not cyclin E1, overexpression induces replication fork stalling through a mechanism dependent on ROS and the replisome-associated ROS sensor PRDX2 (Supplementary Fig. 11b, c; Supplementary Fig. 12b; Supplementary Fig. 13c). Consistently, PRDX2 depletion attenuated R-loop formation in HRASV12-expressing BJ fibroblasts (Supplementary Fig. 12c).

3. In Figure 1, it should be tested whether RNaseH1 expression can suppress DNA damage markers (e.g., γ H2AX). In relation to this, the nature of DNA lesions (e.g., the status of DSBs) has not been scrutinized. What would be the underlying basis of DNA damage that contributes to defective chromosome segregation and mitotic DNA damage after R-loop accumulation?

Response: We tested the effect of HRASV12 or cyclin E1 overexpression on the formation of γ H2AX and 53BP1 foci in different stages of the cells cycle. As mentioned in our response to Reviewer 1 (page 3), we found that none of the oncogenes tested induce extensive DNA damage in S-phase cells (Supplementary Fig. 2). Therefore, we did not perform the suggested RNaseH1 overexpression experiment. Our data suggest that defective chromosome segregation (anaphase bridges and micronuclei) in these cells results from persistent reversed forks as evidenced by rescue of these phenotypes by ZRANB3/SMARCAL1 depletion.

4. The authors uses RNaseH1-dependent rescues of fork progression as sole evidence to support that R-loops are the cause of oncogene-induced replication stress. However, there is no direct evidence of R-loop formation in these oncogene overexpressing cells. This could be validated by using the S9.6 antibody previously used by the lab in Ref#25 either through immunofluorescence or DRIP-seq. A positive control such as PDS or even HU or APH can be included as mentioned by the authors.

Response: To confirm that overexpression HRASV12 or cyclin E1 induces accumulation of R-loops in U2OS cells, we isolated genomic DNA from these cells and performed slot blot using S9.6 antibody, which specifically binds to RNA:DNA hybrids. We observed a significantly higher R-loop signal in samples from oncogene-expressing cells compared to that of cells harboring empty vector. Treating the genomic DNA samples with RNase H before slot blot analysis diminished the S9.6 signals, confirming the specificity of the R-loop signals detected with the S9.6 antibody. These data are now shown in Supplementary Fig. 5 and described on page 7.

5. 5-FU staining suggests that there is an increase in transcriptional activity in G1, S, and G2/M cells that are overexpressing HRASG12V but not in cyclin E1 overexpressing cells, while the authors claim that R-loops seen in S-phase are carried over into mitosis. Can the authors exclude the possibility of R-loop formation within G2/M itself independently of DNA replication?

Response: We consider it unlikely that R-loops detected in G2/M cells were formed independently of DNA replication. If these R-loops resulted from increased transcription, we would expect to see R-loop accumulation in G1 cells as well. However, this is not the case in HRASV12-expressing cells (Fig. 2c). As mentioned in our response to Reviewer 2 (point #1, page 8), in addition to R-loops, we monitored head-on TRCs by measuring colocalization between elongating form of RNAPII and PCNA using PLA. QIBC analysis revealed that overexpression of HRASV12 or cyclin E1 increased RNAPII/PCNA PLA foci in S-phase cells but not in G2 cells (Supplementary Fig. 4; page 7 of the manuscript). These data provide further support for our hypothesis that R-loop formation in HRASV12-/cyclin E1-overexpressing cells is driven by head-on TRCs during S-phase and that a fraction of these R-loops can persist into G2/M, thereby blocking completion of DNA replication.

6. The authors show that ZRANB3 completely prevents oncogene-induced anaphase bridges and micronuclei (Fig. 3). It is an interesting result since other translocases such as SMARCAL1 or HLTF are also known to be involved in fork reversal. These factors should be tested to determine how much individual translocase contributes to the fork reversal process. If solely

dependent on ZRANB3, some mechanistic explanation should be presented considering the way ZRANB3 acts to promote fork reversal.

Response: We tested the effect of SMARCAL1 depletion on replication fork stalling and micronucleation induced by HRASV12 or cyclin E1 overexpression in U2OS cells. We found that SMARCAL1 depletion restored normal fork progression and prevented micronucleation in these cells, as did ZRANB3 depletion. These data are now shown in Supplementary Fig. 7c, d, j.

7. It would be important to test whether PRIMPOL's primase activity is actually required for unrestrained replication fork progression.

Response: We agree that this is an interesting question, but it should be noted that it was not addressed in previous studies on PRIMPOL-mediated repriming (Bai et al., 2020; Quinet et al., 2020; Tirman et al., 2021). All of these studies only used PRIMPOL depletion to assess its role in repriming. Evidence that PRIMPOL-mediated repriming is involved in unrestrained fork progression is provided by S1 nuclease assay demonstrating the presence of ssDNA gaps on DNA fibers. This assay was also used in our study (Fig. 5b). PRIMPOL mediates repriming through its primase and polymerase activities. Although PRIMPOL uses the same active site for both activities, the catalytic core alone is sufficient for polymerase activity but not primase activity. Primase activity requires a zinc finger module located C-terminal to the catalytic core (residues ~372 to 487), which has been shown to be capable of binding to ssDNA (Rechkoblit et al., 2016). Therefore, one could mutate this zinc finger module to generate a PRIMPOL mutant that is defective in primase activity, yet still proficient in polymerase activity. However, to our knowledge, such mutants have not yet been characterized *in vitro*. Therefore, investigating the role of this region of PRIMPOL in repriming would require extensive research, including biochemical characterization of the mutants, generation of stable cell lines expressing PRIMPOL mutants, and phenotypic analyses. This is beyond the scope of our study. Nevertheless, we firmly believe that the data from our S1 nuclease/DNA fiber assays strongly support our conclusion that unrestrained DNA synthesis induced by fork reversal deficiency in HRASV12-/cyclin E1-overexpressing cell involves PRIMPOL-mediated repriming (demonstration of ssDNA gaps).

8. Figure 6 model – there is no experimental evidence supporting that ZRANB3 deficiency rescues the mitotic defect. In relation to this, the physiological relevance of inhibiting fork reversal is not clear. The phenotype of unrestrained DNA synthesis is revealed when cells are forced to lose fork reversal process, and whether unrestrained DNA synthesis plays a meaningful role in normal DNA replication and mitosis is not addressed. Additionally, there is no mechanism of fork remodeling and processing that is uniquely engaged by oncogenes.

Response: Please note that we demonstrate that ZRANB3 depletion prevents the formation of anaphase bridges and micronuclei in HRASV12- or cyclin E1-overexpressing U2OS cell (mitotic defects). As mentioned above these phenotypes are caused by incompletely replicated DNA regions, which impair chromosome segregation in mitosis. If cells do not have ZRANB3, forks stalled at R-loops are restarted during S-phase, which allows for normal chromosome segregation in mitosis and hence prevents chromosomal instability. We have previously shown that in fork reversal-proficient cells, there is an equilibrium between fork reversal and fork restart, which is regulated by RECQ1 and RECQ5 helicases, and that the majority of forks stalled by an R-loop are restarted prior to onset of mitosis (Chappidi et al., 2020). Chromosome segregation errors (anaphase bridges and micronuclei) likely result from

replication fork stalling events in late S-phase, when there is not enough time to restart replication before mitosis. It is conceivable that the same scenario takes place in cells overexpressing oncogenes such as HRASV12 and cyclin E1: majority of R-loop-stalled fork are restarted and those that persist will give rise to anaphase bridges and micronuclei. Please note that we have modified our model to make all these points more clear and also to incorporate our new findings on how HRASV12 induces R-loop-mediated fork stalling (please see Fig. 9).

References

- Andrs, M., Stoy, H., Boleslavskaya, B., Chappidi, N., Kanagaraj, R., Nascakova, Z., Menon, S., Rao, S., Oravetzova, A., Dobrovolna, J., et al. (2023). Excessive reactive oxygen species induce transcription-dependent replication stress. *Nat. Commun.* *14*, 1791.
- Bai, G., Kermi, C., Stoy, H., Schiltz, C.J., Bacal, J., Zaino, A.M., Hadden, M.K., Eichman, B.F., Lopes, M., and Cimprich, K.A. (2020). HLTF Promotes Fork Reversal, Limiting Replication Stress Resistance and Preventing Multiple Mechanisms of Unrestrained DNA Synthesis. *Mol. Cell* *78*, 1237-1251.
- Chappidi, N., Nascakova, Z., Boleslavskaya, B., Porro, A., Lopes, M., Correspondence, P.J.J., Zellweger, R., Isik, E., Andrs, M., Menon, S., et al. (2020). Fork Cleavage-Religation Cycle and Active Transcription Mediate Replication Restart after Fork Stalling at Co-transcriptional R-Loops. *Mol. Cell* *77*, 528-541.
- Hamperl, S., Bocek, M.J., Saldivar, J.C., Swigut, T., and Cimprich, K.A. (2017). Transcription-Replication Conflict Orientation Modulates R-Loop Levels and Activates Distinct DNA Damage Responses. *Cell* *170*, 774-786.
- Kodama, R., Kato, M., Furuta, S., Ueno, S., Zhang, Y., Matsuno, K., Yabe-Nishimura, C., Tanaka, E., and Kamata, T. (2013). ROS-generating oxidases Nox1 and Nox4 contribute to oncogenic Ras-induced premature senescence. *Genes to Cells* *18*, 32-41.
- Kotsantis, P., Silva, L.M., Irmscher, S., Jones, R.M., Folkes, L., Gromak, N., and Petermann, E. (2016). Increased global transcription activity as a mechanism of replication stress in cancer. *Nat. Commun.* *7*, 13087.
- Macheret, M., and Halazonetis, T.D. (2018). Intragenic origins due to short G1 phases underlie oncogene-induced DNA replication stress. *Nature* *555*, 112-116.
- Maya-Mendoza, A., Ostrakova, J., Kosar, M., Hall, A., Duskova, P., Mistrik, M., Merchut-Maya, J.M., Hodny, Z., Bartkova, J., Christensen, C., et al. (2015). Myc and Ras oncogenes engage different energy metabolism programs and evoke distinct patterns of oxidative and DNA replication stress. *Mol. Oncol.* *9*, 601-616.
- Mitsushita, J., Lambeth, J.D., and Kamata, T. (2004). The superoxide-generating oxidase Nox1 is functionally required for Ras oncogene transformation. *Cancer Res.* *64*, 3580-3585.
- Ogrunc, M., Di Micco, R., Liontos, M., Bombardelli, L., Mione, M., Fumagalli, M., Gorgoulis, V.G., and D'Adda Di Fagagna, F. (2014). Oncogene-induced reactive oxygen species fuel hyperproliferation and DNA damage response activation. *Cell Death Differ.* *21*, 998-1012.
- Quinet, A., Tirman, S., Jackson, J., Me, J., Sale, J.E., Vindigni, A., Quinet, A., Tirman, S., and Jackson, J. (2020). PRIMPOL-Mediated Adaptive Response Suppresses Replication Fork Reversal in BRCA-Deficient Cells. *Mol. Cell* *77*, 461-474.
- Rao, S., Andrs, M., Shukla, K., Isik, E., König, C., Bauer, M., Rosano, V., Prokes, J., and Müller, A. (2024). Senataxin RNA/DNA helicase promotes replication restart at co-transcriptional R-

loops to prevent MUS81-dependent fork degradation. *Nucleic Acids Res.* *52*, 10355–10369.

Rechkoblit, O., Gupta, Y.K., Malik, R., Rajashankar, K.R., Johnson, R.E., Prakash, L., Prakash, S., and Aggarwal, A.K. (2016). Structure and mechanism of human PrimPol, a DNA polymerase with primase activity. *Sci Adv.* *2*, e1601317.

Somyajit, K., Gupta, R., Sedlackova, H., Neelsen, K.J., Ochs, F., Rask, M.-B., Choudhary, C., and Lukas, J. (2017). Redox-sensitive alteration of replisome architecture safeguards genome integrity. *Science* *358*, 797–802.

Tirman, S., Quinet, A., Wood, M., Simoneau, A., Zou, L., Vindigni, A., Tirman, S., Quinet, A., Wood, M., Meroni, A., et al. (2021). Temporally distinct post-replicative repair mechanisms fill PRIMPOL-dependent ssDNA gaps in human cells. *Mol. Cell* *81*, 4026-4040.

Weyemi, U., Lagente-Chevallier, O., Boufraquech, M., Prenois, F., Courtin, F., Caillou, B., Talbot, M., Dardalhon, M., Al Ghuzlan, A., Bidart, J.M., et al. (2012). ROS-generating NADPH oxidase NOX4 is a critical mediator in oncogenic H-Ras-induced DNA damage and subsequent senescence. *Oncogene* *31*, 1117–1129.

Ying, S., Minocherhomji, S., Chan, K.L., Palmari-Pallag, T., Chu, W.K., Wass, T., Mankouri, H.W., Liu, Y., and Hickson, I.D. (2013). MUS81 promotes common fragile site expression. *Nat. Cell Biol.* *15*, 1001–1007.

Point-by-point response to the reviewers' comments (NCOMMS-25-21283A)

We would like to thank the reviewers for taking the time to evaluate the revised version of our manuscript and for their positive feedback. We are delighted that all reviewers found the manuscript to be substantially improved.

Answers to specific questions of reviewers:

Reviewer #1 (Remarks to the Author):

The revised manuscript by Oravetzova, Dvorakova et al. includes substantial additional work and improved clarity compared to the initial submission. As noted previously, the study is technically well executed, supported by complementary experimental approaches and appropriate statistical analyses. In the original version, however, the novel mechanistic insights were somewhat limited.

In the revised manuscript, the authors now more effectively develop the concept of mechanistic convergence in oncogene-induced replication stress, focusing on HRASV12 and Cyclin E1. In particular, the clarification that the ROS–PRDX2 axis acts as an upstream driver of replication fork slowing, and that it contributes to the formation of transcription–replication conflicts (TRCs) in HRASV12 but not in Cyclin E1, strengthens the overall contribution of the study. The additional data, including the use of NOX inhibitors, ROS quenching, and PRDX2 depletion experiments, provide useful insight into how distinct oncogenic programs can lead to overlapping replication stress outcomes.

The authors have adequately addressed the specific comments raised during revision, and the revised manuscript presents a large amount of data more systematically and coherently, improving its overall impact for understanding how oncogenes shape replication stress and genome stability. One minor comment from my side is that the authors should consider using a more specific title that better reflects the key data and conclusions presented in the study, rather than relying on a broad and generic title.

Response: We thank Reviewer 1 for the thoughtful assessment of our revised manuscript. We agree that the previous title was somewhat broad, and we have revised it to better reflect the key findings of our study. The new title “*Distinct mechanisms of replication stress induced by oncogenic RAS and cyclin E1 converge on R-loop-dependent fork reversal*” more specifically captures the main conclusions, highlighting the distinct oncogene-induced replication stress mechanisms and the common downstream outcome involving R-loop-mediated fork reversal.

Reviewer #2 (Remarks to the Author):

I appreciate the authors' substantial effort to improve the quality and novelty of the manuscript. The revised version is clearly stronger than the original one, and the inclusion of additional mechanistic data regarding the role of ROS and PRDX2 in the replication stress

induced by HRASV12 has significantly improved the novelty of the study. Nevertheless, I still have some concerns that should be addressed.

Response: We thank Reviewer 2 for the careful evaluation of our revised manuscript and for acknowledging the substantial improvements in quality and novelty compared to the original submission. We are pleased that the reviewer finds the additional data on the role of ROS and PRDX2 in HRASV12-induced replication stress to significantly strengthen the study. We also appreciate the reviewer's continued engagement and constructive feedback. The remaining concerns are addressed below.

The authors conducted new PLA experiments to obtain evidence that overexpression of HRASV12 or cyclin E1 increases the frequency of head-on TRCs. Although they acknowledge that PLA data do not provide direct evidence for the link between TRCs and R-loop, and they have turned down the statements, they claimed that the increase of PLA foci when overexpressing HRASV12 is linked to HO TRCs. They based their conclusions on the observation made by Hamperl et al., 2017 where head-on but not co-directional TRCs increase PLA foci between RNAPII and PCNA. However, these results have been obtained from artificial episomal systems. A recent study has demonstrated that co-directional TRCs, where the RNAPII is which collides with the replication fork traveling ahead, also exists (Bruno et al Mol Cell 2024; doi:10.1016/j.molcel.2023.11.036). These new type of TRCs that should not block replication fork progression can also be detected by these PLA experiments. Consequently, this possibility should be discussed, and it should be acknowledged that at least a fraction of the increase in PLA foci may not correspond to head-on (HO) TRCs. This is especially relevant in the context of HRASV12 overexpression that leads to an increase in transcriptional activity.

Response: As recommended by the reviewer, we now discuss this possibility in the manuscript. The following sentence has been added to page 7: *"However, we cannot exclude the possibility that some of these PLA foci represent co-directional RNAPII collisions behind the replication fork, potentially arising from oncogene-induced fork slowing"*, citing the article by Bruno et al. (2024). Indeed, co-directional transcription resuming on the leading arm of a slowed replication fork (e.g., following ROS-induced TIMLESS-TIPIN dissociation) could bring the RNAPII complex into close proximity with the PCNA clamp and thereby generate a positive signal in RNAPII/PCNA PLA assay.

Moreover, to clearly determine whether HO TRCs are responsible for the observed genomic and mitotic instability, the manuscript would greatly benefit from the inclusion of the HRASV12 overexpression CUT&Tag genomic data the authors are generating and the estimation of replication fork directionality that can be done using public U2OS Repli-seq data. Alternatively, they should be more cautious when assigning directionality to TRCs.

Response: We prefer not to include CUT&Tag data in the current manuscript. To date, we have performed only a single CUT&Tag experiment to map the sites of R-loop formation in cells expressing HRASV12. To draw any conclusions, at least one additional biological replicate would be required, and the analysis would need to be extended to cyclin E1-overexpressing cells. We hope the reviewer will agree that completing this work would require several additional months. While we do plan to perform these experiments, we prefer to include them in a future study, in which we also aim to perform CUT&Tag

experiments to map the location of stalled forks (MCM7, pol epsilon) and transcription complexes (elongating form of RNAPII) in both HRASV12- and cyclin E1-overexpressing cells. We believe that analysis of these datasets will ultimately allow to draw more robust conclusions regarding TRC directionally, R-loop formation and fork stalling upon oncogene activation.

We would like to emphasize that the current manuscript provides experimental evidence demonstrating that oncogene-induced fork stalling and chromosomal instability (anaphase bridges and micronuclei) are caused by R-loops (rescue by RNH1 overexpression). Our hypothesis that these R-loops may result from head-on TRCs is based on previously published findings (e.g., doi: 10.1016/j.cell.2017.07.043) and is not presented as a main conclusion of the manuscript. To avoid any such impression, we have removed the statement “..., likely as a consequence of head-on transcription-replication conflicts (TRCs)” from the abstract.

Reviewer #3 (Remarks to the Author):

The authors addressed many concerns raised in the previous version and strengthened the manuscript. The revised title does not summarize their original findings, though. It is recommended to come up with a more specific title.

Response: We appreciate the reviewer’s feedback and are grateful for his/her recognition of the improvements made to the manuscript. We understand the concern regarding the revised title and agree that it should more accurately summarize our key findings. To address this, we have modified the title to: “*Distinct mechanisms of replication stress induced by oncogenic RAS and cyclin E1 converge on R-loop-dependent fork reversal*”. We believe that this revised title more precisely reflects the core aspects of our study.